# Human and mouse essentiality screens as a resource for disease gene discovery

Pilar Cacheiro [ID] et al.[#]

The identification of causal variants in sequencing studies remains a considerable challenge that can be partially addressed by new gene-specific knowledge. Here, we integrate measures of how essential a gene is to supporting life, as inferred from viability and phenotyping screens performed on knockout mice by the International Mouse Phenotyping Consortium and essentiality screens carried out on human cell lines. We propose a cross-species gene classification across the Full Spectrum of Intolerance to Loss-of-function (FUSIL) and demonstrate that genes in five mutually exclusive FUSIL categories have differing biological properties. Most notably, Mendelian disease genes, particularly those associated with developmental disorders, are highly overrepresented among genes non-essential for cell survival but required for organism development. After screening developmental disorder cases from three independent disease sequencing consortia, we identify potentially pathogenic variants in genes not previously associated with rare diseases. We therefore propose FUSIL as an efficient approach for disease gene discovery.

[#]A full list of authors and their affiliations appears at the end of the paper.

Discovery of the genetic causes of monogenic disorders has shifted from genetic analysis of large cohorts or families with the same phenotype to a genotype-driven approach able to identify ultra-rare variants associated with a disorder in one or few individuals. Published studies by the Centers for Mendelian Genomics (CMG)[1], Deciphering Developmental Disorders (DDD)[2,3] and the Undiagnosed Disease Network[4] have successfully used whole-exome or genome sequencing to find the causal variant in up to 40% of patients. However, the majority of cases remain unsolved. One complimentary approach has been to incorporate gene-level information metrics. These metrics can help to identify candidate variants in genes not previously associated with disease, which are subsequently confirmed as causative in functional in vitro and in vivo studies. Measures of genetic intolerance to functional variation have been used to prioritise candidate disease genes where heterozygous, dominant effects are suspected. These metrics are based on whole-exome and genome sequencing data from broad populations of healthy individuals or cohorts affected by non-severe and non-paediatric disease[5–7]. Use of standardised gene–phenotype associations encoded by the Human Phenotype Ontology (HPO)[8] is another successful strategy that has led to the identification of disease genes by the phenotypic similarity of patients[9]. Phenotype comparison between model organisms and clinical descriptions has also highlighted new candidate gene–disease associations[10]. These successes led us to consider other gene features that could be used to identify human disease genes.

Gene essentiality, or the requirement of a gene for an organism's survival, is known to correlate with intolerance to variation[11] and has been directly assessed in a number of species using high-throughput cellular and animal models[12–14]. Essentiality has been investigated at the cellular level using human cancer cell line screens based on gene-trap, RNAi or CRISPR-Cas9 approaches with the findings that ~10% of protein-coding genes are essential for cell proliferation and survival[15–18]. Project Achilles is extending this approach to characterise 1000 cancer cell lines, importantly correcting for copy number in their essentiality scoring[19,20]. In parallel, the International Mouse Phenotyping Consortium (IMPC), a global research infrastructure that generates and phenotypes knockout (loss of function (LoF)) mouse lines for protein-coding genes, determines viability of homozygotes to assess gene essentiality[10,21,22]. The number of observed homozygous LoF mice generated from an intercross between heterozygous parents allows the categorisation of a gene as lethal (~25% of the genes), subviable (~10%) or viable (~65%)[18,21]. These findings are consistent with results curated from the scientific literature indicating that approximately one-third of protein-coding genes are essential for organism survival[23].

Several levels or definitions of essentiality are to be expected from different approaches[13]. Core essential genes are identified across different model systems, while other essential genes are dependent on context such as culture conditions, tissue or organism developmental stage[24–26]. Quantitative definitions of low and high gene essentiality have been proposed to account for the degree of dependency on external factors, as well as the likelihood of a compensatory mutation rescuing necessary function[26] through mechanisms, such as paralogue buffering[15,27]. Essential genes have also been classified by whether they are known to be associated with human disease, with functional mutations in non-disease-associated genes and with a mouse orthologue that is LoF embryonic lethal suggested as likely to prevent pregnancy, lead to miscarriage or to early death[28]. Other research has reported that orthologues of embryonic lethal LoF mouse genes show an increased association with diseases with high mortality and neurodevelopmental disorders[21,29,30].

Here we provide a new Full Spectrum of Intolerance to Loss-of-function (FUSIL) categorisation that functionally bins human genes by taking advantage of the comprehensive organismal viability screen performed by the IMPC and the cellular viability studies conducted by Project Achilles. We explore the FUSIL categories that span genes from those necessary for cellular survival to genes for which LoF has no detected phenotypic impact on complex organisms, and we demonstrate a strong correlation of genes necessary for mammalian development with genes associated with human disease, especially early-onset, multi-system, autosomal-dominant (AD) disorders. Finally, we describe candidate genes for involvement in AD, developmental disorders where potentially pathogenic variants were identified in unsolved cases from three large-scale exome and genome data sets: the DDD Study[3], the 100,000 Genomes Project (100KGP)[31], and the CMG programmes[1].

## Results

### Functional binning of viability by cross-species comparison.
Human cell-essential genes have been analysed[15–17] in conjunction with lethal genes identified in the mouse[21] by characterising a core set of essential genes at the intersection[13], studying the union of the two[32] or considering them as two separate sets[33]. In these studies, cellular essential genes showed a nearly complete concordance with mouse lethal genes[18,21]. Here we have taken the human orthologue genes for which the IMPC has viability assessments and integrate the gene essentiality characterisation based on the cell proliferation scores determined by the Project Achilles Avana CRISPR-Cas9 screening performed on over 400 cell lines. This identified two sets of lethal genes: a set of cellular lethal genes essential for both a cell and an organism to survive (cellular lethal (CL)), and a set of developmental lethal genes (DL) that are not essential at the cellular level but where LoF is lethal at the organism level. The IMPC viability pipeline also defines genes that result in a subviable (SV) or viable phenotype in LoF mice. We further split the latter into those resulting in an abnormal phenotype (VP, viable with significant phenotype/s) or a normal phenotype (VN, viable with no significant phenotypes detected). As a result, we obtained five mutually exclusive phenotype categories reflecting the FUSIL (Table 1; Supplementary Table 1). The correspondence between viable and SV genes in the mouse and their human orthologues being non-essential in human cell lines was very strong (Table 1, Fig. 1a). An almost complete correspondence was also found between genes essential in human cell lines and mouse genes that are lethal in LoF strains. However, while 35% of genes lethal in the mouse were essential in human cell lines (CL bin), the remaining 65% have not been identified as cell essential and were classified as essential for organism development (DL bin). A near identical pattern was observed when other cellular essentiality data sets were used from previously published studies[15–17], with most genes ending up in the same category (96% overlap; Supplementary Fig. 1c, Supplementary Table 2).

### Biological process and pathway analysis.
An enrichment analysis of Gene Ontology (GO) biological processes (BPs) showed that the two lethal FUSIL categories (CL and DL) were involved in different types of biological processes (Fig. 1b, Supplementary Data 1). Whereas the set of CL genes was enriched for nuclear processes (DNA repair, chromosome organisation, regulation of nuclear division), the DL genes were enriched in morphogenesis and development functions (embryo development, appendage development, tissue morphogenesis, specification of symmetry). In contrast, genes in the SV and viable categories (VP, VN) were not significantly enriched in any biological process despite

**Table 1 FUSIL categories.**

| Mouse category | Human cell line category | Number of genes | % Overlap | FUSIL category |
|---|---|---|---|---|
| Lethal | Essential | 413 | 35.09% | Cellular lethal (CL) |
| Lethal | Non-essential | 764 | 64.91% | Developmental lethal (DL) |
| Subviable | Essential | 16 | 3.66% | — |
| Subviable | Non-essential | 421 | 96.34% | Subviable (SV) |
| Viable with phenotypic abnormalities | Essential | 18 | 0.95% | — |
| Viable with phenotypic abnormalities | Non-essential | 1867 | 99.05% | Viable with phenotype (VP) |
| Viable with normal phenotype | Essential | 2 | 0.62% | — |
| Viable with normal phenotype | Non-essential | 318 | 99.38% | Viable with no phenotype (VN) |

Integration of data from human cell essentiality screens from the Avana data set and mouse phenotypes from IMPC screens for 4446 protein-coding genes that have data in both resources and a high-quality orthologue. This defined five mutually exclusive categories of intolerance to loss of function and the number of human protein-coding genes is shown for each. For 627 of the viable mouse lines, the number of procedures with QCed data available was <50% and thus they were classified as Viable with insufficient procedures (see "Methods", Supplementary Table 1) and not incorporated into these FUSIL categories. The Viable with phenotype (VP) category indicates that the phenotypes of the knockout (loss of function) mouse line differ significantly from the wild-type mice in at least one of the many parameters measured as part of the IMPC phenotyping pipeline (average of 163 parameters measured on any given mouse).

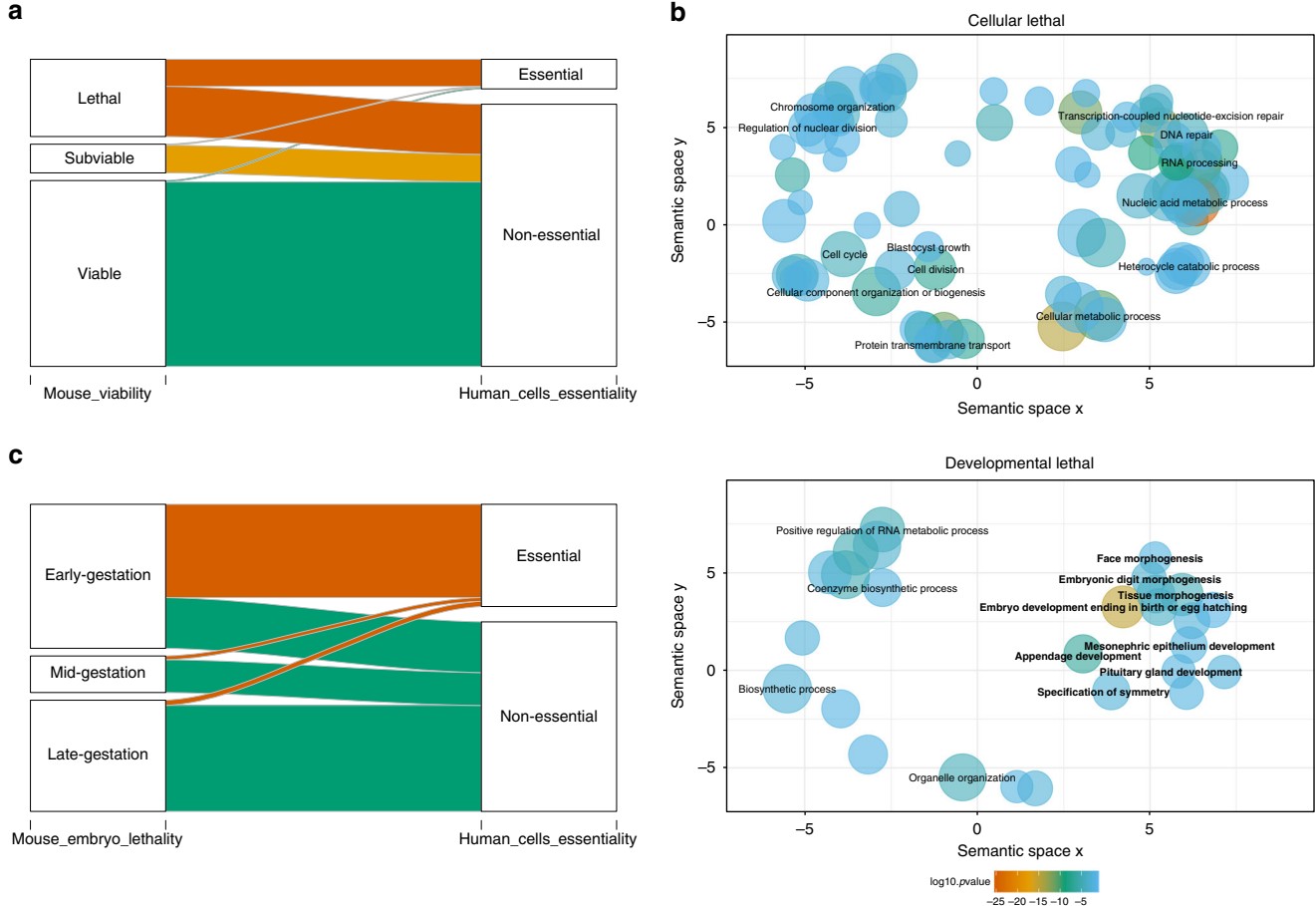

**Fig. 1 Cross-species FUSIL categories of intolerance to LoF. a** Correspondence between primary viability outcomes in mice and human cell line screens. The sankey diagram shows how human orthologues of mouse genes with IMPC primary viability assessment (lethal, subviable and viable) regroup into essential and non-essential human cell categories; the width of the bands is proportional to the number of genes. **b** Gene Ontology Biological Process (GO BP) enrichment results. Significantly enriched GO terms at the biological process level were computed using the set of IMPC mouse-to-human orthologues incorporated into the FUSIL categories as a reference (Table 1) and identified after correcting for multiple comparisons. Significant results were only found for the cellular and developmental lethal gene categories. Bubble size is proportional to the frequency of the term in the database and the colour indicates significance level as obtained in the enrichment analysis. The GO terms associated with embryo development are in bold. **c** Correspondence between mouse embryonic lethality stage and essentiality in human cell lines. Embryonic lethal LoF strains are assessed for viability at selected stages during embryonic development: early (gestation) lethal (prior to E9.5), mid (gestation) lethal (E9.5–E14.5/15.5), late (gestation) lethal (E14.5/E15.5 onwards). E embryonic day.

reasonable sample sizes, probably indicating diverse roles for these genes. A pathway analysis showed consistent results, with genes belonging to cell cycle and DNA-associated pathways significantly overrepresented among CL genes. Developmental biology pathways, on the other hand, were only found enriched among DL genes (Supplementary Fig 2, Supplementary Data 2). GO BP analysis showed numerous enriched RNA-processing terms mainly in the CL category. Related processes involving RNA polymerase and transcription were nominally enriched in the DL fraction, reflecting how transcriptional regulation is central to organism development. Taken together, these analyses provide additional evidence to make a distinction between the two sets of lethal genes in terms of their biological function.

**Cellular essential genes correlate with early lethal genes**. For those genes found to be lethal, the IMPC performed a secondary viability assessment of null homozygous embryos to determine the time of embryonic death. Four hundred lethal genes were classified in one of the three windows of lethality: early gestation (49.25%), mid-gestation (12.50%), and late gestation lethal genes (38.25%), confirming previous findings that nearly half of lethal mouse genes with an embryonic screening correspond to embryos which die prior to embryonic day 9.5 (E9.5)[21]. When this information was combined with the human cell data set, we observed a strong concordance between the stage of embryo lethality and essentiality at the cellular level: 65% of early gestation lethal genes are essential in human cell lines, whereas only 10% of mid-gestation lethal genes and <5% of late gestation lethal genes fall into this category (Fig. 1c, Supplementary Table 3).

**FUSIL categories and associated gene features**. In humans, essential genes have been shown to be located in regions with lower recombination rates[34]. We observe a trend of increasing recombination rate from most to least essential, with CL genes representing a clearly distinct category and significant differences between bins for most pairwise comparisons (Fig. 2a, Supplementary Table 4). Consistent with higher expression values previously associated with essential genes[15], we show a decreasing trend in human gene expression levels from most to least essential FUSIL bins, as measured by median GTEx expression across a wide range of tissues and cell lines (Fig. 2b). Similar continuous trends were observed for other gene features previously associated with essential genes[15,28,35], including protein–protein interaction network properties (Fig. 2c) or the likelihood of the gene product being part of a protein complex (Fig. 2d). CL genes also stand out as a singular category regarding the number of paralogues (Fig. 2e).

In contrast, features associated with mutational rate appear to peak in the DL or SV bins: probability of mutation based on gene context (Fig. 2f), transcript length (Fig. 2g), and strength of negative selection measured by Gene-level Integrated Metric of negative Selection (GIMS) scores (Fig. 2h). However, these observed increases in the DL and SV genes relative to CL were only statistically significant for transcript length (Supplementary Table 4). A similar effect is observed with several intolerance to variation scores from the Genome Aggregation Database (gnomAD). Higher (but not statistically significant) probability of LoF intolerance (pLI) values were observed in the DL and SV genes relative to CL, with a lower frequency of pLI values close to 0 among the viable bins when compared to the CL, DL and SV sets (Fig. 2i, Supplementary Fig. 3, Supplementary Table 5). A reverse trend, given the nature of this metric, was detected for the observed/expected (o/e) LoF scores (upper bound fraction) (Fig. 2j).

**Developmental lethal bin is enriched in human disease genes**. Previous studies have reported associations between disease and essential genes using different criteria and data sets[29,36]. Our first report on developmental phenotypes showed a significant enrichment for disease genes in the IMPC essential genes[21]. By segmenting this set of genes into three mutually exclusive categories (CL, DL and SV; Fig. 3a), we found that while the CL and SV fractions showed a moderate enrichment for disease genes compared to all other categories (odds ratios (ORs) > 1) and the two bins containing viable genes (VP and VN) were significantly depleted (ORs < 1), the highest overrepresentation of Mendelian disease genes was found in the DL fraction (2.6-fold increased odds). This finding was consistent with an early study defining a set of peripheral essential genes[25] and a recent review comparing cellular and mouse essential genes[33].

Analysis of the mode of inheritance of the disease genes in each bin showed the CL fraction had the lowest proportion of AD disorders, while the DL and VN fractions showed higher proportions (uncorrected $P$ values of 0.0143 and 0.004, respectively, Fisher's exact test) (Fig. 3b). The latter, however, was based on a relatively small number of disease genes (40) with AD/autosomal-recessive (AR) annotations in this category relative to 119 and 286 seen in the CL and DL fractions, respectively. The proportion of known haploinsufficient genes among disease genes was greatest in the DL fraction with twice as many as in the CL fraction (Fig. 3c).

We analysed the FUSIL categories for gnomAD's o/e LoF scores (upper bound fraction), which are used to evaluate a gene's tolerance to LoF variation. Our results showed that essential genes are more intolerant to inactivation compared to viable genes and that the DL and SV categories showed the peak intolerance although the difference with respect to the set of CL genes was not significant (Fig. 2j, Supplementary Table 5). Again, this is consistent with an overrepresentation of AD disorders among DL genes. Similar results were found when we investigated other intolerance scores (Supplementary Fig. 3, Supplementary Table 5).

The proportion of disease genes associated with an early age of onset (antenatal/prenatal and neonatal) was highest in the CL and DL gene sets, with the percentage of later-onset associated genes increasing as we move toward more viable categories (Fig. 3d). The degree of pleiotropy, measured by the number of physiological systems affected according to HPO annotations, followed a similar pattern, with CL and DL genes showing the highest number of affected systems (Fig. 3e).

In summary, our results suggest that disease genes in the DL fraction correlated with earlier age of onset, multiple affected systems and AD disorders (Supplementary Table 6). Given this, we compared the genes in our FUSIL bins with the genes identified as likely to be causative of developmental disorders in the DDD Study as reported in the Development Disorder Genotype–Phenotype Database (DDG2P) and found a strong enrichment for these genes in the DL FUSIL bin (3.9-fold increase; Fig. 3f). The DDDG2P resource has a larger representation of non-consanguineous patients with de novo mutations compared to the OMIM and Orphanet resources[37], so we next compared enrichment of DL genes by mode of inheritance across the three resources and found the strongest enrichment for monoallelic disease genes in the DDDG2P resource (Fig. 3g, h).

These findings were corroborated when we explored diagnostic-grade genes that have a high level of evidence for disease associations as curated by experts for Genomics England (Supplementary Fig. 4). The DL category showed an even higher enrichment in developmental disease-associated genes for a subset of highly confident genes from DDG2P[3] (4.4-fold increase; Supplementary Fig. 4e) and as did a set of genes associated with

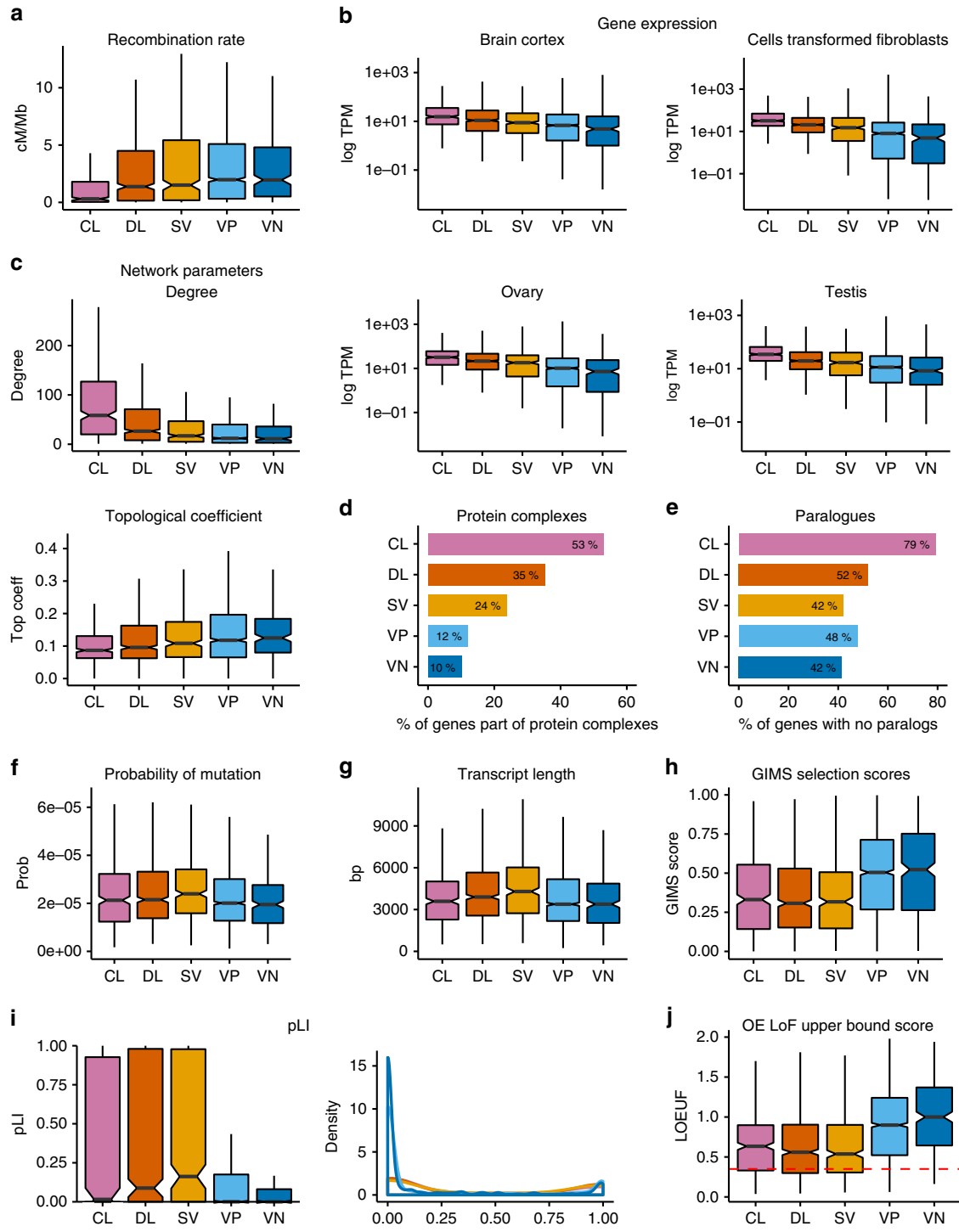

foetal anomalies from the PAGE Consortium[38] (4.9-fold increase; Supplementary Fig. 4f).

These results are particularly relevant for the identification of new disease genes using these FUSIL bins: for Mendelian genes, despite the enrichment in the DL fraction, the highest proportion was still found in the much larger VP bin (Fig. 3a, g), but for developmental disorders, as represented by DDG2P, the DL fraction contained the majority of genes, reaching a percentage close to 50% for monoallelic disease genes (Fig. 3g, h), and up to 57% when different curated subsets of genes involved in developmental disorders and foetal anomalies were investigated

(Supplementary Fig. 4g, h). Thus, we attempted to identify strong, novel candidate genes for undiagnosed cases of AD, development disorders by extracting 163 genes from the DL bin genes ($n = 764$), which had the following properties: (i) not described as associated with human disease by OMIM, Orphanet or DDG2P and (ii) highly intolerant to a LoF mutation (pLI > 0.90 or o/e LoF upper bound < 0.35 or haploinsufficiency (HI) score < 10). These 163 prioritised DL genes are more likely to belong to the same protein family or biological pathway of known monoallelic developmental disease genes and more frequently interact with them when compared to genes across the FUSIL bins not

**Fig. 2 FUSIL categories and human gene features. a** Notched box plots showing the distribution of recombination rates for the different FUSIL bins. Human recombination rates[58] were mapped to the closest gene and average recombination rates per gene were computed. **b** Distribution of human gene expression values for different tissues. Median logTPM expression values from the GTEx database for selected non-correlated tissues are shown. **c** Protein–protein interaction network parameters. Notched box plots showing the distribution of degree and topological coefficient computed from human protein–protein interaction data extracted from STRING. Only high-confidence interactions, defined as those with a combined score of >0.7, were kept. **d** Protein complexes. Bar plots representing the percentage of genes in each FUSIL bin being part of a protein complex (human protein complexes). **e** Paralogues. The bar plot shows the percentage of genes without a protein-coding paralogue gene in each FUSIL bin. Paralogues of human genes were obtained from Ensembl Genes 95. A cut-off of 30% amino acid similarity was used. **f** Probability of mutation. Distribution of gene-specific probabilities of mutation from Samocha et al.[65]. **g** Transcript length. Maximum transcript lengths among all the associated gene transcripts (Ensembl Genes 95, hsapiens data set). **h** GIMS Selection Score. Distribution of Gene-level Integrated Metric of negative Selection (GIMS)[66] scores across the different FUSIL bins. **i** Probability of loss-of-function intolerance (pLI) retrieved from gnomAD2.1. Notched box plots and density plots showing the bimodal distribution of this score, with higher values indicating more intolerance to variation. **j** Distribution of gnomAD o/e LoF scores. Upper bound fraction of the confidence interval around the observed versus expected LoF score ratio (gnomAD 2.1.). A score <0.35 (dashed line) has been suggested to identify intolerant to LoF variation genes[56]. For **a**–**c**, **f**, **g**–**j**: centre line, median; notch, CI around the median; box edges, interquartile range, 75th and 25th percentile, respectively; whiskers, 1.5 times the interquartile range; outliers not shown. Significance for pairwise comparisons for all features is shown in Supplementary Tables 4 and 5. CL cellular lethal (pink), DL developmental lethal (orange), SV subviable (yellow), VP viable with phenotypic abnormalities (light blue), VN viable with normal phenotype (dark blue).

associated with disease (Supplementary Fig. 5, Supplementary Data 3). Only the set of non-disease CL genes showed similar results regarding pathways and interactors.

**Screening of unsolved developmental disorder cases.** We next focussed on unsolved diagnostic cases from three large rare disease sequencing programmes to investigate potential disease candidates within our set of 163 prioritised DL genes (Supplementary Data 4).

First, DDD makes publicly available a set of functional de novo variants and/or rare homozygous, hemizygous or compound heterozygous LoF research variants of unknown significance found in genes that are not associated with human disease according to OMIM and DDG2P. These variants were found in 4293 children with developmental disorders who participated in the UK DDD study and remained undiagnosed. Given recent findings from DDD that most developmental disorder cases could be explained by de novo coding mutations[37], we searched for heterozygous, de novo variants in this data set that affected any of the prioritised 163 DL candidate genes described above and found variants in 44 genes that met these criteria. Second, we searched 18,000 rare disease cases from the 100KGP[31] and discovered de novo variants in undiagnosed patients with intellectual disability in 47 of the 163 genes in our candidate set. Lastly, the CMG, a collaborative network of centres to discover new genes responsible for Mendelian phenotypes provided a list of phenotypes studied and potential associated genes[39], which included around 2000 genes. A set of 14 genes overlapping with the 163 DL candidates are classified as either Tier 1 or Tier 2 genes (see "Methods").

We additionally explored denovo-db, a database containing de novo variants identified in the literature[40], and annotated our set of 163 prioritised genes with this information. We found at least one functional de novo variant reported in the database for 108 (66%) genes, of which 83 (51%) contained entries collected from resources other than DDD[2] (Supplementary Data 4).

There was some degree of overlap between the candidates identified in the three programmes (Fig. 4a) and for the next stage we focussed on genes with evidence from both the 100KGP (where we had detailed patient phenotypes and variant information) and either DDD (variants and high-level phenotypes available) or the CMG (gene and high-level phenotypes available). We particularly focussed on 9 genes where the associated variants were not present in any population in gnomAD and each gene was also not highly tolerant to missense variation (o/e missense < 0.8; Fig. 4b, Supplementary Data 4). For these

genes, further evidence for candidacy was gathered based on the number of unrelated families and phenotypic similarities between them, protein–protein interactions with known developmental disorder genes, embryonic and adult mouse gene expression in relevant tissues and embryonic and adult mouse phenotypes that recapitulate the clinical phenotypes. Here we present two examples, *VPS4A* and *TMEM63B*, where the patient phenotype and genetic evidence is compelling as well as showing pheno-copying in the mouse where the IMPC has produced the first knockout lines for these genes. For the other 7 genes (*ATP6V0A1*, *MAEA*, *CMIP*, *PKN2*, *SPTBN1*, *RBMS1* and *PUM2*), there was functional data supporting the association, but the patient evidence is currently less strong as we only have single, intellectual disability cases with detailed clinical phenotypes, or de novo variants are also observed in cases affected with other types of rare disease.

*VPS4A* (HGNC:13488, vacuolar protein sorting 4 homologue A) had no previously reported pathogenic variants and is highly intolerant to LoF and missense variants (gnomAD v.2.1., pLI = 0.928, o/e LoF = 0.139, o/e missense = 0.532). De novo variants in *VPS4A* were detected in two unsolved, 100KGP intellectual disability cases but not in any of the other 18,000 cases representing most types of rare disease. These variants are not observed in gnomAD and both patients exhibited consistent intellectual disability, developmental delay, delayed motor development, microcephaly and eye abnormalities, including cataracts (Supplementary Table 7, Supplementary Fig. 6). In addition, a Tier 2 CMG candidate was described with similar phenotypes of microcephaly, epilepsy, frontoencephalocele, right spastic hemi-paresis and psychosocial retardation (Supplementary Table 7, Supplementary Fig. 6). The IMPC's data for the first mouse knockout of the orthologous *Vps4a* gene indicated preweaning lethality of the homozygotes and, in the case of the heterozygotes, abnormal skin morphology, enlarged spleen and lens opacity, potentially modelling the eye phenotypes seen in the patients. LacZ staining in E12.5 embryos showed widespread expression. While the LoF mutants are lethal at P14, secondary viability of *Vps4a* mutants shows that they are viable at E18.5 (6/30, 20%) but display gross abnormalities at manual observation and by micro-computed tomography (microCT) imaging. Homozygous *Vps4a* E18.5 embryos are smaller than wild types (WTs), with abnormal body curvature, omphalocele, small and compressed heart, abnormal spinal cord curvature and abnormal brain development. Within the brain, microCT images showed evidence of abnormalities in the thalamus, thinning of the midbrain and a smaller cerebellum and pons compared to the WT littermates.

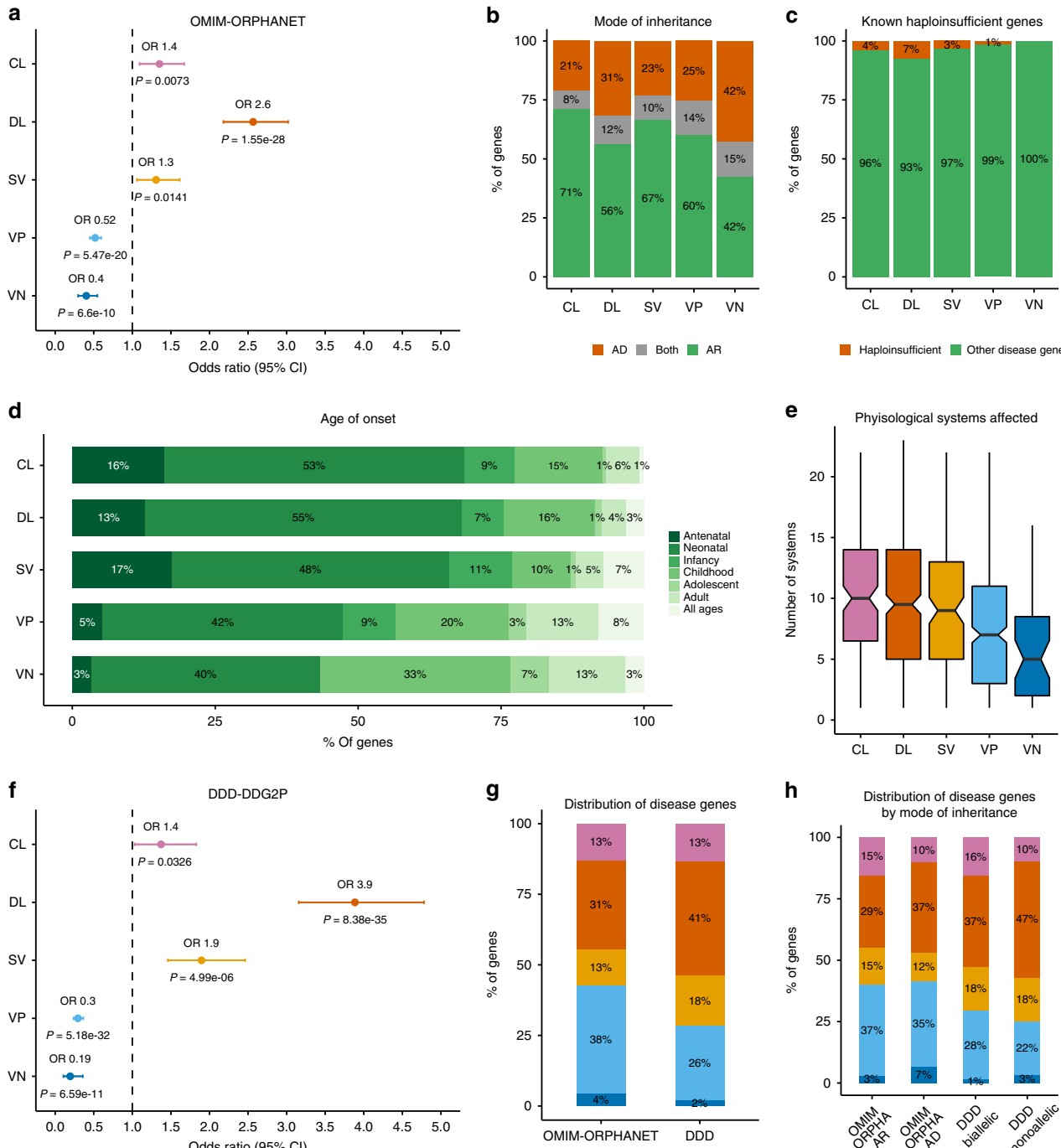

The volume changes of the midbrain/cerebellum/pons might also be related to the enlargement of the fourth ventricle (Fig. 4c). Interestingly, *VPS4A* is known to directly interact in humans with an intellectual disability gene, *CHMP1A*, from nuclear magnetic resonance, affinity chromatography, pull down and two hybrid assays, and both are part of the necroptosis and endocytosis pathways[41]. Variants in *CHMP1A* cause pontocerebellar hypoplasia type 8 (OMIM:614961), with similar phenotypes to patients with *VPS4A* variants: severe psychomotor retardation, pontocerebellar hypoplasia, decreased cerebral white matter, thin corpus callosum, abnormal movements, hypotonia, spasticity, and variable visual defects. High gene expression levels are present across all tissues as measured in humans (GTEx data), both disease and non-disease related. Similarly, high levels of

expression were also seen in WT mouse embryos from 4 to 36 somites according to Deciphering the Mechanism of Developmental Disorders (DMDD)[42]. In addition, we found two de novo variants reported in denovo-db database[40]: a nonsense (stop-gained) variant with an associated intellectual disability phenotype and a missense variant in a patient with autism. None of these variants are present in gnomAD.

*TMEM63B* (HGNC:17735, transmembrane protein 63B) is also extremely intolerant to LoF and missense variants (gnomaD v. 2.1., pLI = 1.00, o/e LoF = 0.07, o/e missense = 0.475) but has no previously reported pathogenic variants. De novo variants in unsolved, developmental disorder cases were identified in 1 DDD case and 4 unrelated 100KGP participants with intellectual disability but none of the other 18,000 100KGP cases

**Fig. 3 Human disease genes and FUSIL bins. a** Enrichment analysis of Mendelian disease genes. Combined OMIM-ORPHANET data was used to compute the number of disease genes in each FUSIL bin. Odds ratios were calculated by unconditional maximum likelihood estimation (Wald) and confidence intervals (CIs) using the normal approximation, with the corresponding adjusted P values for Fisher's exact test. **b** Distribution of disease-associated genes according to mode of inheritance. Disease genes with annotations regarding the mode of inheritance according to the Human Phenotype Ontology[8]. **c** Haploinsufficient genes. Known haploinsufficient genes curated by ClinGen (percentage with respect to the total number of disease genes in each bin). **d** Age of onset as described in rare diseases epidemiological data from Orphanet (Orphadata). The earliest age of onset associated with each gene was used. Bar plots representing the percentage of disease genes associated with each age of onset for each FUSIL category. **e** Distribution of the number of physiological systems affected. The phenotypes (HPO) associated with each gene were mapped to the top level of the ontology to compute the number of unique physiological systems affected. **f** Enrichment analysis of developmental disorder genes. The Developmental Disorders Genotype-Phenotype Database (DDD-DDG2P) set of genes was used to compute the number of developmental disorder genes in each FUSIL bin. These genes were compared against non-disease genes (OMIM, ORPHANET and DDD-DD2GP). Odds ratios were calculated by unconditional maximum likelihood estimation (Wald) and confidence intervals (CIs) using the normal approximation, with the corresponding adjusted P values for Fisher's exact test. **g** Distribution of disease genes. Percentage of distribution of Mendelian and developmental disorder genes among the different FUSIL categories. **h** Distribution of disease genes by mode of inheritance. Percentage of distribution of Mendelian and developmental disorder genes among the different FUSIL categories according to the mode of inheritance reported in the HPO (set of Mendelian disease genes) and DDD (developmental disease-associated genes). CL cellular lethal (pink), DL developmental lethal (orange), SV subviable (yellow), VP viable with phenotypic abnormalities (light blue), VN viable with normal phenotype (dark blue), DDD/DDD-DDG2P Deciphering Developmental Disorders database of genes that are likely causative of developmental disorders. For **e**, centre line, median; notch, CI around the median; box edges, interquartile range, 75th and 25th percentile, respectively; whiskers, 1.5 times the interquartile range.

(Supplementary Table 8). These variants are not observed in gnomAD and the exact same variant (ENST00000259746: c.130G>A) was detected in three of the families, causing a p.44Val>Met change in a transmembrane helix that is predicted to be pathogenic. The clinical data available from the four 100KGP cases showed consistent intellectual disability and abnormal movement and brain morphology phenotypes, with seizures also observed in three of the patients (Supplementary Table 8, Supplementary Fig. 7). For the DDD case, the high-level phenotypes available were consistent with the 100KGP cases. IMPC data for the first mouse knockout of the orthologous *Tmem63b* gene showed preweaning lethality of the homozygotes and, in the case of the heterozygotes, abnormal behaviour, hyperactivity and limb-grasping phenotypes that are consistent with the human patients. Expression analysis placed *TMEM63B* in a cluster of genes that are expressed at medium levels from early to late development. High levels of gene expression in disease-related tissues, particularly for the brain cerebellum and muscular–skeletal tissues, were identified in humans (GTEx data). High levels of expression were also seen across all mouse embryonic developmental stages according to DMDD, with GXD data and the IMPC's mouse embryo *lacZ* annotation supporting neuronal expression during development (Fig. 4d).

## Discussion

The diagnostic rate of large-scale, rare disease sequencing programmes ranges from 20% to 40%[31,43], leaving the majority of patients without a diagnosis and with the associated personal, psycho-social and healthcare cost this entails. New disease-associated gene discovery methods are needed to complement current sequencing approaches[44]. Here we demonstrate that the FUSIL categorisation of gene essentiality, combined with intolerance to variation scores, patient phenotypes and their overlap with those observed in mouse lines with null alleles can assist in the prioritisation of disease–causal variant candidates. We show an enrichment for disease genes among developmental lethal genes, which is consistent with the proposed model where disease-associated genes occupy an intermediate position between highly essential and non-essential genes[28,45]. In this model, highly essential genes will not be associated with human diseases because any function-altering mutation will likely lead to miscarriage or early embryonic death. The set of cellular lethal genes is indeed enriched for pathways associated with developmental disorders, but the minimal enrichment in developmental disease-associated genes may be explained by embryonic lethality at a very early

stage of development[33]. Our results provide further evidence that highly essential genes needed for cellular processes are less likely to be associated with disease than developmental essential genes, suggesting a new complementary approach for finding such disease genes and understanding disease mechanisms.

An interesting finding was the dichotomy of trends observed for gene-associated features (Fig. 2). We replicated previously observed trends where genetic features are most differentiated between the two ends of the FUSIL spectrum, e.g. genes with paralogues, gene expression, number of protein–protein interactions or the likelihood of being part of a protein complex[15,28,35]. The CL bin showed the lowest rates of recombination, and both the CL and DL fractions exhibited significantly lower rates than the viable categories. This is consistent with previous findings that genes with essential cell functions, and showing accumulation of disease-associated mutations, concentrate in genomic regions with suppressed recombination[34]. The strong enrichment of CL genes for the presence in protein complexes and a lack of paralogues would suggest that these genes should be particularly intolerant to damaging mutations with no functional compensation to buffer critical cell processes[15,46]. For other gene features associated with mutational rate, the trends peak in the DL and SV bins, leading to the counter-intuitive observation that CL genes are less constrained against variation than DL genes; however, the differences between these two categories are non-significant for the different constraint scores evaluated. The significant enrichment for longer transcript lengths and Mendelian disease genes aligns with previous observations that disease genes tend to be longer[47], and genetic variant intolerance metrics such as shet, pLI and o/e LoF are highly dependent on gene length[7,13].

Recent analysis from the DDD study estimates that de novo heterozygous variants will be the cause of nearly half of all undiagnosed, non-consanguineous individuals with developmental disorders[37]. Since these early-onset disorders are particularly overrepresented among developmental lethal genes, we focussed on a set of unsolved developmental disorder cases from DDD, the 100KGP and CMG. We identified 82 disease candidates among the 163 highly LoF intolerant genes from the DL fraction not previously associated with Mendelian disease. Nine genes in particular were observed in multiple consortia and are based on de novo variants in unsolved cases of developmental disorders that have never been observed in the general population. Further exploration of overlapping genotypes and phenotypes between the patients and with embryonic and adult mouse phenotype evidence from the IMPC, alongside evidence from expression and

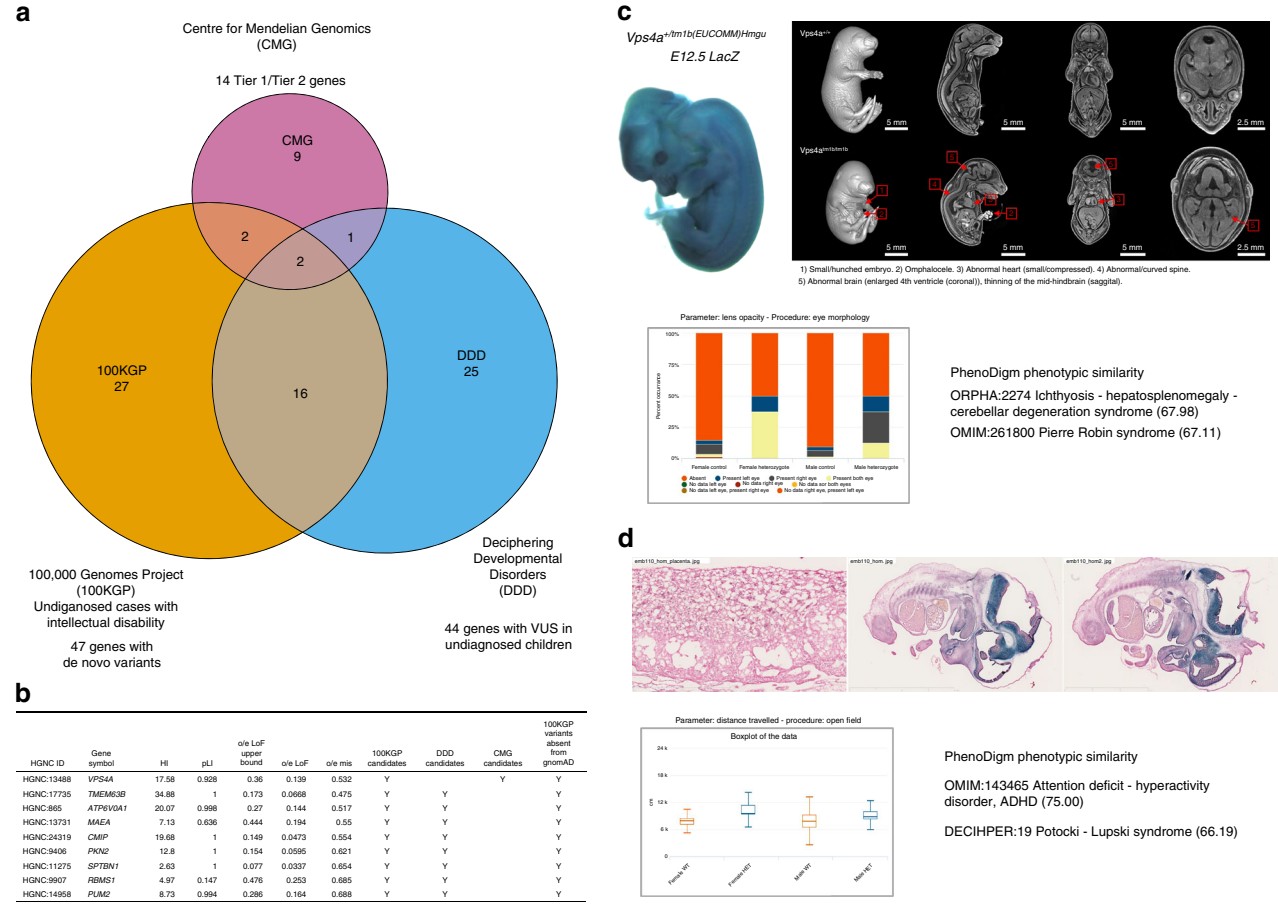

**Fig. 4 Developmental disorders gene candidate prioritisation. a** Venn diagram showing the overlap between DL prioritised genes with evidence from 3 large-scale sequencing programmes. Overlap between the set of 163 developmental genes highly intolerant to LoF variation (pLI > 0.90 or o/e LoF upper bound < 0.35 or HI < 10) and not yet associated with disease and the set of candidate genes from three large rare disease sequencing consortia: 100KGP, CMG, and DDD. **b** Set of nine candidate genes. The selected genes met the following criteria: (1) evidence from both the 100KGP (with detailed clinical phenotypes and variants) and either DDD (variants and high-level phenotypes available) or CMG (gene and high-level phenotypes available), (2) the associated variants were not present in gnomAD, and (3) intolerance to missense variation; these genes were further prioritised based on the number of unrelated probands and the phenotypic similarity between them and the existence of a mouse knockout line with embryo and adult phenotypes that mimic the clinical phenotypes. **c** Mouse evidence for *VPS4A*. IMPC embryonic phenotyping of homozygous mutants at E18.5 showed abnormal/curved spine and abnormal brain among other relevant phenotypes. The phenotypic abnormalities observed in heterozygous knockout mice include lens opacity. Heterozygous mouse phenotypic similarity to known disorders as computed by the PhenoDigm algorithm. **d** Mouse evidence for *TMEM63B*. IMPC homozygous mouse embryo lacZ imaging at E14.5 supporting neuronal expression during development. Heterozygous IMPC knockout mice associated phenotypes included abnormal behaviour evaluated through different parameters. The heterozygous mice showed a high phenotypic similarity with several developmental disorder phenotypes. VUS variant of unknown significance.

protein–protein interactions with known developmental disorder genes, further supported these associations. Two genes were particularly compelling disease candidates from a clinical perspective, as gene variants were only observed in patients with intellectual disability belonging to multiple, unrelated families with specific phenotypes in common. One de novo variant in *TMEM63B* was observed in three DDD and 100KGP patients with consistent intellectual disability, movement and brain morphology phenotypes that were recapitulated in the IMPC *Tmem63b* mutant mouse. Concordant phenotypes were also observed between 3 patients with *VPS4A* variants from the CMG and 100KGP programs and the E18.5 mutant mouse embryos from the IMPC. Future identification of other families with similar variants segregating with disease and functional characterisation of the specific human variants will be required to establish a definitive role for these genes in disease. The IMPC partners are already supporting the CMG, 100KGP, other disease sequencing projects such as KidsFirst and the wider rare disease

community through CRISPR/Cas9 production and phenotyping of mouse lines modelling patient-specific, potentially pathogenic variants.

Cross-species data integration is not without its limitations. Not all human genes have a clear one-to-one orthologue in the mouse genome. In addition, a significant proportion of the mouse protein-coding genome is not yet phenotyped by the IMPC and is lacking viability data. In this investigation, we chose to focus on high-quality, IMPC viability calls based on robust statistical methodologies and not to integrate the numerous, additional data from literature curation of mice with knockout alleles because the variation in methods, genetic context and gene targeting approach are known to affect embryonic lethality. Indeed, we observed a ~10% disagreement between embryonic lethality published in the literature on a variety of genetic backgrounds versus the IMPC observations made exclusively on an inbred C57BL/6N background ("Methods" and Supplementary Fig. 8). Human cell line data also has caveats, including the haploid and

immortalised nature of some of the cell lines. The gene constraint scores based on human sequencing data primarily identify selection against heterozygous variation and may fail to detect short genes, recessively acting genes or HI status[48], while the homozygous viability screens in knockout mice typically measure recessive effects. Hence, a moderate overlap between the set of essential genes identified through the different approaches is to be expected[13]. Further investigation into haploinsufficient, essential genes in the mouse is more technically challenging but may reveal further disease candidates. Combining approaches to compute intolerance to LoF, e.g. integrating FUSIL bins with other constraint scores, improves the ability to identify disease-associated genes compared to the performance of standalone metrics (Supplementary Fig. 9). Moreover, although in the present study we have targeted a particular category of disorders, an additional feature offered by the FUSIL framework is the ability to utilise all the bins to focus on other types of disease, including those associated with late-onset, less severe phenotypes or recessive mode of inheritance. Alternative approaches integrating different strategies of gene prioritisation are expected to follow, acting as an accelerator in Mendelian disease-associated gene discovery.

In summary, this study highlights how the information on gene essentiality may be used to prioritise potential pathogenic variants in unknown disease genes from human sequencing studies. Clinical researchers assessing candidate disease genes should consider using high-quality model organism data in conjunction with gene constraint scores from human sequencing projects. We intend to incorporate such data into our Exomiser variant prioritisation tool[49], which together with patient phenotypes, may facilitate the genetic diagnosis by prioritising genes in the relevant FUSIL category. Large-scale projects such as the IMPC, the gnomAD resource and Project Achilles are continuing to generate larger data sets, making the resources richer and more robust for these analyses. Future work will explore the mechanisms behind how these redefined essential categories correlate to other functional attributes and, ultimately, the evolutionary constraints imposed upon gene essentiality.

## Methods

**Main data sets**. In this study, we examined genes with viability data for null homozygote mice produced by the IMPC (https://www.mousephenotype.org). We obtained high-to-moderate-confidence human orthologues and integrated selected human genetic data. In particular, we incorporated gene viability data obtained in human cell screens (initially 11 cell lines corresponding to three studies)[18]; in this study, we used the Project Achilles Avana data set comprising 485 cell lines (release 18Q3 of August 2018), which is part of the Cancer Dependency Map (DepMap) project[19]. We also incorporated constraint scores (shet, Residual Variation Intolerance Score (RVIS), pLI and o/e LoF), disease information (OMIM, ORPHANET, HPO, DDD) and DMDD annotations (see below for details). We used HGNC and Mouse Genome Informatics (MGI) as stable identifiers in our analysis to avoid problems associated with gene symbol changes (synonyms and previous symbols, which may lead to ambiguous identifiers) [https://www.genenames.org/download/statistics-and-files/; HGNC protein-coding gene file, downloaded 18.11.14].

**IMPC primary and embryo viability assessment**. We conducted an analysis on 4934 genes with primary adult viability data currently available in IMPC Data Release (DR) 9.1. This release included the 1751 genes analysed in Dickinson et al.[21] and the 4237 genes analysed in Munoz-Fuentes et al.[18], corresponding to DR4.0 and DR7.0, respectively, and additional data collected since then.

Viability data generated by the IMPC was analysed as defined in IMPRESS (the International Mouse Phenotyping Resource of Standardised Screens, https://www.mousephenotype.org/impress/). A minimum of 28 pups were genotyped before weaning, and the absence of knockout (null) homozygote pups would classify the gene as lethal. Thus lethal lines are defined as those with an absence of live null homozygous pups, while subviable lines are those with <12.5% live homozygous pups (half of the 25% expected; $P < 0.05$, binomial distribution). Viable mouse lines are those for which homozygous (null and WT) and heterozygous pups are observed in normal Mendelian ratios. A viable call was also made when there were <28 total pups and homozygous null pups ≥ 4 (as this would result in ≥14% homozygous pups when 28 pups were genotyped). We filtered out genes for which sample size was insufficient (total pups < 28, $n = 1$ gene) and hemizygous genes

($n = 13$ genes), as well as those with conflicting calls (genes that appear in more than one viability category, $n = 34$ genes). The resulting set comprises 4886 genes, of which 1171 had a lethal phenotype, 449 a subviable phenotype, and 3266 a viable phenotype (24%, 9% and 67%, respectively).

The IMPC also implements a dedicated embryonic pipeline for lethal lines, in which null homozygous embryo viability is assessed at selected stages during embryonic development, including E9.5, E12.5, E14.5–E15.5 and E18.5. Viability at a given stage is assessed by scoring homozygous embryos for the presence of a heartbeat at dissection, as described in Dickinson et al.[21]. To establish windows of lethality, we considered a gene lethal at a given stage if no live homozygous embryos were identified after scoring at least 28 live embryos and viable if any live homozygous embryos were identified, irrespective of additional phenotype features. Following Dickinson et al.[21], we defined windows of lethality as "prior to E9.5", "E9.5–E12.5", "E12.5–E14.5/15.5", "E15.5–E18.5", "after E14.5/E15.5", and "after E18.5". Lines with incomplete data to define these windows were excluded. For this study, windows were combined to yield three functional groups: early (gestation) lethal (prior to E9.5), intermediate (mid-gestation) lethal (E9.5–E12.5, E12.5–E14.5/E15.5), and late (gestation) lethal (E14.5/E15.5–E18.5, after E14.5/E15.5 and after E18.5). Out of 523 for which secondary screen data are available, 400 could be assigned to one of the described windows. Additional time points are required to complete the window assignment for the remaining 123 genes (Supplementary Table 3).

**Orthologue mapping**. Orthologues were obtained using the HCOP tool developed by HGNC, based on 12 established inference methods (https://www.genenames.org/tools/hcop/; HCOP file with human and mouse orthologue inferences downloaded 18.10.31). We determined the confidence on the orthologue prediction based on the number of methods that supported each inference (12–9 methods, 100–75%, good-confidence orthologue; 8–5, 67–42%, moderate-confidence orthologue; 4–1, 33–8%, low-confidence orthologue; 0 no orthologue). We kept orthologues for which at least one gene, the human or the mouse gene, was protein coding and the orthologue inference score was ≥5. Of those, we kept genes for which the score was maximum in both directions, mouse to human and human to mouse (and also filtered out genes with duplicated maximum scores). This resulted in 4664 genes. Of these, 33 genes with IMPC conflicting phenotypes and 17 genes with no adult viability data and insufficient embryo data to call a viability phenotype were not considered further. Among the remaining 4614 genes, 1185 (26%) genes had a lethal phenotype, 443 (9%) a subviable phenotype, and 2986 (65%) a viable phenotype. Of these, 4446 genes had human cell viability data (Avana data set; see next section).

**Essentiality in human cells**. In our previous study[18], we used viability data as reported for 11 cell lines from 3 studies[15–17]. Here we used the Project Achilles Avana data set and CRISPR-Cas9 proliferation (essentiality) scores, which is part of the Cancer Dependency Map (DepMap) project[21]. This data set comprises viability data on 17,634 genes in 485 cell lines (https://depmap.org/portal/achilles/; release 18Q3 in August 2018), with lower values indicating more intolerance (i.e. more essential). The Avana data set presents several advantages as compared to the previous studies. It is a larger data set, viability is measured using the same units for all cell lines (allowing us to obtain a mean per gene) and the data are corrected for copy number variation[20]. Gene identifiers were provided as Entrez identifiers (NCBI) and converted to HGNC identifiers.

**Determining functional bins**. For each gene, we obtained a mean of the Avana proliferation scores and integrated with our data set of mouse genes with viability data based on the mouse-to-human orthologues (obtained as described above). We observed that, for genes with an Avana mean score < −0.45, the mouse null homozygotes were lethal in almost all cases, while genes with an Avana mean score >−0.45 presented lethal, subviable or viable phenotypes (Supplementary Fig. 1a, Supplementary Fig. 1b). A similar pattern was observed when a different resource for cell essentiality—based on 11 cell lines from 3 different studies—was used (methods and threshold criteria explained in Munoz-Fuentes et al.[18] (Supplementary Fig. 1c).

F1 scores were derived from confusion matrices generated when considering different Avana mean scores and the classification from the previous studies, and a mean score cut-off of −0.45 was found to maximise the F1 scores across the different data sets (Supplementary Fig. 1d, Table 1, Supplementary Table 1). We therefore set a threshold where genes with a mean Avana ≤ −0.45 were considered essential in cell lines, while the set >−0.45 corresponded to cellular non-essential genes. Based on all this evidence, we categorised two sets of genes, Cellular Lethal (CL) and Developmental Lethal (DL) genes, comprising 413 and 764 genes, respectively (Table 1).

Genes with a subviable or viable phenotype were classified for their human orthologue almost in all cases as non-essential (Avana mean score for each gene >−0.45), except for 16 and 22 genes for the subviable and viable categories, respectively, which had a mean Avana scores below but very close to the −0.45 threshold. These genes were non-essential based on data from the three human cell studies. Thus we distinguish a Subviable group (SV) with Avana mean score >−0.45, as well as two outlier groups, SV.outlier and V.outlier, with Avana

mean score ≤ −0.45. We classified viable genes further, into those with at least one IMPC significant phenotype (VP) or no IMPC significant phenotypes (VN). A number of viable genes ($n = 627$) had <50% phenotype procedures obtained so far (the IMPC releases are snapshots of ongoing characterisation for many lines), and thus we termed them V.insuffProcedures (Supplementary Table 1).

These categories, established using the Avana cell scores, were almost identical (96% concordance) to those established using a gene essentiality classification (essential/non-essential) based on the 11 cell lines from 3 studies[18].

All the subsequent analysis focussed on the main five FUSIL bins: CL, DL, SV, VP, and VN. Sankey diagrams representing the mappings between mouse and human cell essentiality categories were plotted with the R package *alluvial*[50]. GO BP enrichment was conducted with the R package *category*[51], with the entire set of IMPC genes as the reference set. The Benjamin and Hochberg (BH) method was applied for multiple testing correction[52] and an adjusted $P$ value < 0.05 was considered significant. The results were plotted using the REVIGO algorithm for semantic similarity and redundancy reduction[53]. The algorithm selects representative GO terms out of the significant results, maximising semantic representation and enriched/statistical significance (settings: SimRel semantic similarity measure, medium similarity, *Homo sapiens* database). Significant results were only found for the CL and DL gene categories. In Fig. 1b, the bubble size is proportional to the frequency of the term in the database and the colour indicates the significance level as obtained in the enrichment analysis, after correcting for multiple testing.

Reactome pathway enrichment analysis was performed by means of the R/Bioconductor package *ReactomePA*[54]. The same approach for multiple testing correction followed in the GO analysis was applied.

**Previous knowledge on mouse viability.** The IMPC is an ongoing project. Given that about ¼ of the mouse genome has been screened for viability to date, we decided to compare our results with previous annotations from MGI[55]. We found a total of 4599 mouse genes with embryo lethality annotations (50 Mammalian Phenotype Ontology terms as described in Dickinson et al.[21]) once conditional mutations were excluded from a total of 13,086 genes with any phenotypic annotation (including normal phenotype). That would mean that approximately 35% of the genes for which a knockout mouse has been reported in the literature and curated by the MGI showed some type of embryo/perinatal lethality. As the MGI resource also includes IMPC data, we subsequently excluded those gene–phenotype associations corresponding to IMPC alleles. This results in 9254 mouse genes with phenotypic annotations other than IMPC with a PubMed ID (old papers with no abstract available, conference abstracts and direct data submissions were therefore excluded for this analysis). For 3362 (36%) of these genes, we found annotations related to embryonic lethality.

Even though the procedures for determining viability may differ from the standardised viability protocol followed by the IMPC, this percentage is very close to that found using IMPC data, with 26% of the screened genes lethal and 10% of the screened genes subviable. For 2115 mouse genes with both IMPC and non-IMPC phenotypic annotations available to infer viability, we found discrepancies for a set of 63 genes that were scored as lethal by the IMPC and with no previous records of lethality in MGI, as well as 154 genes scored as viable by the IMPC and with some type of lethality annotation reported in MGI (Supplementary Fig. 8, Supplementary Table 9; Mouse Genome Database at the Mouse Genome Informatics website [http://www.informatics.jax.org; MGI_GenePheno.rpt; data accessed 19.02.06]).

**Constraint scores based on human population data.** The RVIS[5] was downloaded from http://genic-intolerance.org/; version CCDSr20, with lower values indicating more intolerance to variation. Estimates of selection against heterozygous protein-truncating variants (shet) were obtained from the supplementary material of Cassa et al.[11] with higher values indicating more intolerant to variation. The probability of a gene being intolerant to LoF (pLI)[6], with higher values indicating more intolerance, the o/e ratio of LoF with the corresponding upper bounds (LOEUF) the (o/e) ratio of missense and the (o/e) ratio of synonymous alleles, with lower values indicating more intolerance to variation, were retrieved from gnomAD2.1 (https://gnomad.broadinstitute.org/)[7,56]. The HI was obtained from the DDD consortium (https://decipher.sanger.ac.uk/about#downloads/data; downloaded BED file, 18.11.27, Haploinsufficiency Predictions Version 3)[57]. High ranks (e.g. 0–10%) indicate a gene is more likely to exhibit HI while low ranks (e.g. 90–100%) indicate a gene is more likely to not exhibit HI. In all cases, gene identifiers were obtained as symbols and converted to HGNC IDs using the multi-symbol checker provided by HGNC (https://www.genenames.org/tools/multi-symbol-checker/).

A comparative analysis of FUSIL categories with gnomAD pLI scores was conducted considering a threshold of pLI > 0.90 to define highly constrained genes, i.e. LoF intolerant genes (Supplementary Fig. 9).

**Recombination rates.** We used the average genetic map computed from the paternal and maternal human genetic maps from Halldorsson et al.[58] (Data S3). The recombination rate (cMperMb) was provided for genomic intervals, and we mapped each interval to the closest protein coding gene—Ensembl 96 GRCh 38—(upstream or downstream) by means of R_bedtools_closest function in *HelloRanges* library[59]. Gene positions were obtained through *biomaRt*[60]. Once we

assigned the recombination rates for the intervals provided to a certain gene, average recombination rates per gene were computed.

**Gene expression.** Human gene median transcripts per kilobase million (TPM) values by tissue were downloaded from the GTEx portal [https://gtexportal.org/home/datasets; accessed 18.12.02, file "GTEx_Analysis_2016-01-15_v7_RNA-SeQCv1.1.8_gene_median_tpm.gct.gz"][61]. Gene symbols were mapped to HGNC ID identifiers. Spearman correlation coefficients between gene expression values across tissues were estimated, only TPM values for relevant (non-correlated) tissues were considered for further analysis.

**Protein–protein interaction data.** Human protein–protein interaction data were downloaded from STRING website [accessed 18.11.18]. Only high-confidence interactions, defined as those with a combined score >0.7 were used for further analysis[62]. Ensembl protein IDs were mapped to HGNC IDs using Ensembl biomaRt (Ensembl Genes 94)[60]. Several network parameters for every node in the network were computed with Cytoscape (v3.5.1) plug-in Network Analyzer[63]. Spearman correlation coefficients between different network parameters were estimated; only non-correlated parameters were considered for further analysis.

**Protein complexes.** A core set of human protein complexes was downloaded from Corum 3.0[64] (http://mips.helmholtz-muenchen.de/corum/#download; accessed 18.12.03). Gene symbols were mapped to unambiguous HGNC ID identifiers corresponding to protein-coding genes.

**Paralogues.** Paralogues were retrieved with biomaRt (https://www.ensembl.org/index.html;Ensembl; Genes 95 version [data accessed 19.02.18])[60]. Only genes with HGNC IDs were considered, and only those protein-coding paralogues with an HGNC ID were kept for downstream analysis. A cut-off of 30 for the percentage of identical amino acids in the paralogue compared with the gene of interest was used for the computation.

**Probability of mutation.** Per-gene probabilities of mutations (all types of mutations) were obtained from Samocha et al.[65]. Gene symbols were mapped to HGNC IDs (Probabilities are shown as 10^all).

**Transcript length.** Transcript lengths were retrieved with biomaRt R package (Ensembl Genes 96, hsapiens_gene_ensembl data set)[60]. For each HGNC ID, the maximum transcript lengths was computed from all the gene transcripts.

**Selection scores.** GIMS scores, which combine multiple comparative genomics and population genetics to measure the strength of negative selection, were obtained from Sampson et al.[66]. (Table_S1). Gene symbols were mapped to unambiguous HGNC ID identifiers.

**Gene–disease associations.** Disease-associated genes curated by OMIM (https://omim.org/) and Orphanet (https://www.orpha.net) were analysed through our PhenoDigm pipeline[67] [Data accessed 19.01.16]. DECIPHER developmental disorders genes are defined as those reported to be associated with developmental disorders, compiled by clinicians as part of the DDD study to facilitate clinical feedback of likely causal variants. The DDG2P[3] is categorised into the level of certainty that the gene causes developmental disease (confirmed or probable), the consequence of a mutation (LoF, activating, etc.) and the allelic status associated with disease (monoallelic, biallelic, etc.) [DDG2P version: DDG2P_19_2_2019.csv; https://decipher.sanger.ac.uk/ddd#ddgenes].

Given that OMIM contains a certain number of genes involved in susceptibility to multifactorial disorders and other non-Mendelian gene–disease associations and that DDD-DDG2P also includes probable and possible gene–disease associations, we decided to investigate additional sets of curated gene–disease associations. In particular, we explored the gene panels curated by Genomics England and incorporated in its PanelAPP. Only those genes categorised as "green", i.e. there is a high level of evidence for the gene–disease association, and therefore are considered as diagnostic grade, were explored. Five different sets of genes were analysed: one corresponding to the total set of "green" genes included in any Genomics England gene panel (PanelAPP, 285 panels with at least one gene classified as green) and those genes belonging to the following panels: DDG2P (Developmental disorders, set of "green" genes from DDG2P panel, which contains a subset of DDG2P genes with one of the following levels of evidence: Confirmed or both DD and IF), PD (set of "green" genes from the Paediatric disorders gene panel), ID (set of "green" genes from the Intellectual disability gene panel) and FA (set of "green" genes from the foetal anomalies panel, which contains a subset of genes associated with developmental disorders developed by the PAGE study (Prenatal Assessment of Genomes and Exomes) with a confirmed disease confidence rating that underwent additional review and curation) [https://panelapp.genomicsengland.co.uk; data accessed 19.06.02].

**Mode of inheritance and physiological systems affected.** Mode of inheritance and number of physiological systems affected were annotated for each gene

according to Human Phenotype Ontology annotations [https://hpo.jax.org/app/download/annotation; downloaded 19.02.19]. The file ALL_SOURCES_ALL_-FREQUENCIES_genes_to_phenotype.txt provides a link between genes and HPO terms[8]. AR inheritance (HP:0000007) and AD inheritance (HP:0000006) annotations were selected for downstream analysis. Phenotype ontology terms associated with each gene were mapped to the top level of the HPO to compute the number of unique physiological systems affected.

An additional set of haploinsufficient genes from ClinGen[68] (https://www.clinicalgenome.org/) ($n = 295$) was used for the analysis [https://github.com/macarthur-lab/gene_lists; data accessed 19.02.19].

Genomics England PanelAPP also contain information about the mode of inheritance. We restricted the analysis to those genes associated with one of the following modes of inheritance: monoallelic, biallelic, or both with consistent annotation across different panels [https://panelapp.genomicsengland.co.uk; data accessed 19.06.02].

**Age of onset**. The age of onset was obtained from rare diseases epidemiological data (Orphadata) [http://www.orphadata.org/cgi-bin/epidemio.html; data accessed 19.02.13]. The earliest age of onset associated with each gene was selected for downstream analysis.

**Disease gene enrichment**. For each FUSIL bin, ORs were computed from a contingency table with the number of disease and non-disease genes for each one of the categories versus the remaining set of IMPC genes with FUSIL information. For each one of the disease categories that were analysed, the corresponding subset of disease-associated genes was compared to the overall set of non-disease genes. ORs were calculated by unconditional maximum likelihood estimation (Wald) and confidence intervals using the normal approximation, with the corresponding two-sided $P$ values for the test of independence calculated using Fisher's exact test (adjusted $P$ values, BH adjustment[52]).

**Candidate genes annotation and prioritisation strategy**. A gene set consisting of those developmental lethal genes ($n = 764$) that were not associated with a Mendelian disorder according to OMIM, ORPHANET or DECIPHER ($n = 387$) and highly likely to be haploinsufficient (HI % < 10 | o/e lof upper bound < 0.35 | pLI > 0.90) ($n = 163$) was used to identify candidate genes for undiagnosed cases of developmental disorders with heterozygous mutations.

Further analysis was conducted to compare our set of 163 prioritised DL genes with those genes non-associated with disease from the remaining FUSIL bins as well as with the entire set of genes in each FUSIL bin. We focussed on evaluating our disease candidate genes against a set of genes associated with developmental disorders with a monoallelic mode of inheritance reported: the set of DDD monoallelic "green" genes as curated in Genomics England PanelAPP ($n = 291$). We used PFAM protein family annotations (biomaRt, Ensembl Genes 97 version, hsapiens_gene_ensembl data set [data accessed 19.07.31])[60], pathways from Reactome[69] (lowest-level pathways, *Homo sapiens*) [https://reactome.org/download/current/Ensembl2Reactome.txt; data accessed 19.07.31] and protein interactors from STRING[62] as explained above (STRING ppI annotations with a combined score >0.7). We compared our selected set of DL genes with those in the different FUSIL bins that are not associated with disease and with the entire set of genes in each FUSIL bin. For each one of the three features (protein families, pathways and interactors), we computed the percentage of genes in each category sharing a PFAM protein family, a Reactome pathway (lowest level) or directly interacting with any monoallelic developmental disease gene.

Deciphering developmental disorders (DDD) research variants (found in ~2000 genes) were downloaded from https://decipher.sanger.ac.uk/ddd#research-variants [accessed 18.11.05]. This is a set of variants of unknown significance found in 4293 children with developmental disorders who participated in the UK DDD study. It includes functional de novo and rare LoF homozygous, compound heterozygous and hemizygous variants in genes that are not associated with developmental disorders and are not reported in OMIM in children who remain undiagnosed in the DDD study.

CMG[1] Tier 1 and Tier 2 level genes (~2000 genes) were obtained from http://mendelian.org/phenotypes-genes [accessed 19.28.02]. No information about individual variants is provided.

De novo variants (<0.1% in the 100KGP and public resources such as gnomAD and predicted to have effect on the coding region of genes) in undiagnosed patients with intellectual disability from the 100KGP and associated clinical phenotypes (HPO terms) were extracted by querying the data available in the GeCIP research environment [accessed 18.10.20] and intersecting with our set of 163 prioritised genes.

Denovo-db database, a resource containing human de novo variants identified in the literature[40] was also explored. Mainly focussed but not limited to neurodevelopmental phenotypes and congenital heart disease, it also includes published de novo variants associated with other phenotypes, e.g. autism, Tourette syndrome or schizophrenia or controls) [denovo-db, Seattle, WA URL: denovo-db.gs.washington.edu; data accessed 19.07.24]. Only those variants with a predicted effect on the coding region were considered for the purpose of this study (synonymous variants were excluded).

Research candidates were identified in 82 of the 163 prioritised genes (highly LoF intolerant not previously been associated with Mendelian disease). Out of the total number of 47 genes with heterozygous de novo variants from undiagnosed cases in the 100KGP project and with extensive clinical phenotype available, 19 overlap with genes with heterozygous de novo variants from the DDD set of research candidates and 4 of them with Tier 1 or Tier 2 genes from CMG (of which 2 were shared between the 3 data sets). We next focussed on this set of overlapping genes to narrow down the search for strong candidates, and we discarded those genes where the variants were present at any frequency in gnomAD along with those intolerant to missense variation (gnomAD o/e missense score < 0.8). This resulted in 9 genes that were then prioritised based on the presence of unrelated probands with phenotypic similarities and the existence of knockout mice—with embryonic and/or adult phenotypes—mimicking the clinical phenotypes.

For the set of candidate genes, expression analysis was conducted using BrainSpan[70] and FUMA Gene2Func[71] with all protein-coding genes as a background. Other mouse annotation resources include: DMDD (https://dmdd.org.uk/), GXD resource (www.informatics.jax.org/expression.shtml) and IMPC mouse embryo lacZ imaging (http://www.mousephenotype.org/data/imageComparator?¶meter_stable_id=IMPC_ELZ_063_001&acc=MGI:2387609).

The phenotypic similarity between the IMPC mouse models and known disorders was computed by the PheneDigm algorithm[67].

**Software**. Data analysis and visualisation was performed in R[72].

**Ethics statement**. Mouse production, breeding and phenotyping at each centre was done in compliance with each centre's ethical animal care and use guidelines in addition to the applicable licensing and accrediting bodies, in accordance with the national legislation under which each centre operates (Animal Welfare and Ethical Review Body (AWERB), UC Davis Institutional Animal Care and Use Committee (IACUC), Animal Care Committee (ACC) of the Toronto Centre for Phenogenomics, IACUC of MARC, The RIKEN Tsukuba Animal Experiments Committee, Com'Eth. agreement nb: 17, Regierung von Oberbayern, The Jackson Laboratory Institutional Animal Care and Use Committee (IACUC), Institutional Animal Care and Usage Committee). All efforts were made to minimise suffering by considerate housing and husbandry. All phenotyping procedures were examined for potential refinements disseminated throughout the IMPC. Animal welfare was assessed routinely for all mice.

All patient data used in this study was either accessed through the public websites provided by DDD and the CMG or, in the case of the 100KGP, through the research environment provided by Genomics England and conforming to their procedures. All participants in the 100KGP have provided written consent to provide access to their anonymised clinical and genomic data for research purposes. The 100KGP research and clinical project model and its informed consent process has been approved by the National Research Ethics Service Research Ethics Committee for East of England – Cambridge South Research Ethics Committee.

**Reporting summary**. Further information on research design is available in the Nature Research Reporting Summary linked to this article.

## Data availability

The data that support the findings of this study are available in the BioStudies database under accession number S-BSST293. All other data supporting the findings of the study are included in the paper or its supplementary information tables. Additional information on mouse phenotypes is available through the IMPC web portal (https://www.mousephenotype.org/) and the IMPC FTP repository (ftp://ftp.ebi.ac.uk/pub/databases/impc/).

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

## Acknowledgements

This work was supported by NIH grant U54 HG006370. IMPC-related mouse production and phenotyping was funded by the Government of Canada through Genome Canada and Ontario Genomics (OGI-051) for NorCOMM2 (to C.M.) and the National Institutes of Health and OD, NCRR, NIDDK and NHLBI for KOMP and KOMP2 Projects U42 OD011175 and UM1OD023221 (to C.M., K.C.K.L.), Infrafrontier grant 01KX1012, EU Horizon2020: IPAD-MD funding 653961 (to M.H.d.A.); EUCOMM: LSHM-CT-2005-018931, EUCOMMTOOLS: FP7-HEALTH-F4-2010-261492 (to W.G.W.); UM1 HG006348; U42 OD011174; U54 HG005348 (to A.L.B.), NIH U54706HG006364 (to A.L.B.); Wellcome Trust grants WT098051 and WT206194 (to D.J.A.); The French National Centre for Scientific Research (CNRS), the French National Institute of Health and Medical Research (INSERM), the University of Strasbourg and the "Centre Europeen de Recherche en Biomedicine", and the French state funds through the "Agence Nationale de la Recherche" under the frame programme Investissements d'Avenir labelled (ANR-10-IDEX-0002-02, ANR-10-LABX-0030-INRT, ANR-10-INBS-07 PHENOMIN to J.H.). This research was made possible through access to the data and findings generated by the 100,000 Genomes Project. The 100,000 Genomes Project is managed by Genomics England Limited (a wholly owned company of the Department of Health). The 100,000 Genomes Project is funded by the National Institute for Health Research and NHS England. The Wellcome Trust, Cancer Research UK and the Medical Research Council have also funded research infrastructure. The 100,000 Genomes Project uses data provided by patients and collected by the National Health Service as part of their care and support. We are also grateful for the data access provided by the DDD and CMG projects. The DDD study presents independent research commissioned by the Health Innovation Challenge Fund [grant number HICF-1009-003], a parallel funding partnership between Wellcome and the Department of Health, and the Wellcome Sanger Institute [grant number WT098051]. The views expressed in this publication are those of the author(s) and not necessarily those of Wellcome or the Department of Health. The study has UK Research Ethics Committee approval (10/H0305/83 granted by the Cambridge South REC and GEN/284/12 granted by the Republic of Ireland REC). The research team acknowledges the support of the National Institute for Health Research, through the Comprehensive Clinical Research Network. The Centers for Mendelian Genomics are funded by the National Human Genome Research Institute, the National Heart, Lung, and Blood Institute and the National Eye Institute. Broad Institute (UM1 HG008900), Johns Hopkins University School of Medicine/Baylor College of Medicine (UM1 HG006542), University of Washington (UM1 HG006493) and Yale University (UM1 HG006504).

## Author contributions

P.C., V.M.F., T.F.M. and D.S. contributed to the data analysis, writing of the paper and design and execution of the work. S.A.M., M.E.D., M.B. and K.A.P. contributed to data analysis and review of the manuscript. H.M., H.W., D.G.L. and J.R.S. contributed to data acquisition and data handling. T.K. and H.H.M. contributed to development of the software and databases and review of the manuscript. J.P., V.N., T.S., C.-W.S., A.C., C.J.L., H.W.-J., L.T., H.C., M.S. and T.H. contributed to mouse phenotyping and data acquisition. H.F. led the mouse phenotyping. L.M.J.N. led the production of mouse models and cohorts for phenotyping and contributed to review of manuscript. A.M.F. led the mouse and embryo phenotyping and data acquisition and contributed to review of manuscript. V.G.-D. led the mouse phenotyping and contributed to review of manuscript. D.S., T.F.M., A.L.B., J.D.H., M.H.d.A., W.W., Y.H., D.J.A., R.S., F.M., S.W., R.E.B., H.P., S.D.M.B., C.M., G.T.-V., A.-M.M. and K.C.K.L. are PIs of the key programmes who contributed to the management and execution of the work and contributed to review of the manuscript. The additional IMPC consortium members all contributed to data acquisition and data handling.

## Competing interests

The authors declare no competing interests.

## Additional information

Pilar Cacheiro[1,25], Violeta Muñoz-Fuentes[2,25], Stephen A. Murray[3], Mary E. Dickinson[4,5], Maja Bucan[6], Lauryl M.J. Nutter[7], Kevin A. Peterson[3], Hamed Haselimashhadi[2], Ann M. Flenniken[8], Hugh Morgan[9], Henrik Westerberg[9], Tomasz Konopka[1], Chih-Wei Hsu[4], Audrey Christiansen[4], Denise G. Lanza[5], Arthur L. Beaudet[5], Jason D. Heaney[5], Helmut Fuchs[10], Valerie Gailus-Durner[10], Tania Sorg[11], Jan Prochazka[12], Vendula Novosadova[12], Christopher J. Lelliott[13], Hannah Wardle-Jones[13], Sara Wells[9], Lydia Teboul[9], Heather Cater[9], Michelle Stewart[9], Tertius Hough[9], Wolfgang Wurst[14,15,16], Radislav Sedlacek[12], David J. Adams[13], John R. Seavitt[5], Glauco Tocchini-Valentini[17], Fabio Mammano[17], Robert E. Braun[3], Colin McKerlie[7,18], Yann Herault[19], Martin Hrabě de Angelis[10,20,21], Ann-Marie Mallon[9], K.C. Kent Lloyd[22], Steve D.M. Brown[9], Helen Parkinson[2], Terrence F. Meehan[2,25] &

Damian Smedley [1,25]*, The Genomics England Research Consortium, The International Mouse Phenotyping Consortium

[1]Clinical Pharmacology, William Harvey Research Institute, School of Medicine and Dentistry, Queen Mary University of London, London EC1M 6BQ, UK. [2]European Molecular Biology Laboratory, European Bioinformatics Institute (EMBL-EBI), Wellcome Genome Campus, Hinxton, Cambridge CB10 1SD, UK. [3]The Jackson Laboratory, Bar Harbor, ME 4609, USA. [4]Departments of Molecular Physiology and Biophysics, Baylor College of Medicine, Houston, TX 77030, USA. [5]Departments of Molecular and Human Genetics, Baylor College of Medicine, Houston, TX 77030, USA. [6]Department of Genetics, Perelman School of Medicine, University of Pennsylvania, Philadelphia, PA 19104, USA. [7]The Centre for Phenogenomics, The Hospital for Sick Children, Toronto, ON M5T 3H7, Canada. [8]The Centre for Phenogenomics, Lunenfeld-Tanenbaum Research Institute, Mount Sinai Hospital, Toronto, ON M5T 3H7, Canada. [9]Medical Research Council Harwell Institute (Mammalian Genetics Unit and Mary Lyon Centre), Harwell, Oxfordshire OX11 0RD, UK. [10]German Mouse Clinic, Institute of Experimental Genetics, Helmholtz Zentrum München, German Research Center for Environmental Health, 85764 Neuherberg, Germany. [11]Université de Strasbourg, CNRS, INSERM, Institut Clinique de la Souris, PHENOMIN-ICS, 67404 Illkirch, France. [12]Czech Centre for Phenogenomics, Institute of Molecular Genetics of the Czech Academy of Sciences, Vestec, 252 50 Prague, Czech Republic. [13]Wellcome Trust Sanger Institute, Hinxton, Cambridge CB10 1SA, UK. [14]Institute of Developmental Genetics, Helmholtz Zentrum München, German Research Center for Environmental Health GmbH, 85764 Neuherberg, Germany. [15]Department of Developmental Genetics, Center of Life and Food Sciences Weihenstephan, Technische Universität München, 85764 Neuherberg, Germany. [16]Deutsches Institut für Neurodegenerative Erkrankungen (DZNE) Site Munich, Munich Cluster for Systems Neurology (SyNergy), Adolf-Butenandt-Institut, Ludwig-Maximilians-Universität München, 80336 Munich, Germany. [17]Monterotondo Mouse Clinic, Italian National Research Council (CNR), Institute of Cell Biology and Neurobiology, 00015 Monterotondo Scalo, Italy. [18]Translational Medicine, The Hospital for Sick Children, Toronto, ON M5T 3H7, Canada. [19]Université de Strasbourg, CNRS, INSERM, Institut de Génétique, Biologie Moléculaire et Cellulaire, Institut Clinique de la Souris, IGBMC, PHENOMIN-ICS, 67404 Illkirch, France. [20]Department of Experimental Genetics, Center of Life and Food Sciences Weihenstephan, Technische Universität München, 85354 Freising-Weihenstephan, Germany. [21]German Center for Diabetes Research (DZD), 85764 Neuherberg, Germany. [22]Mouse Biology Program, University of California, Davis, CA 95618, USA. [25]These authors contributed equally: Pilar Cacheiro, Violeta Muñoz-Fuentes, Terrence F. Meehan, Damian Smedley. Members of The Genomics England Research Consortium are listed at the end of the paper. Members of The International Mouse Phenotyping Consortium are listed at the end of the paper. *email: d.smedley@qmul.ac.uk

## The Genomics England Research Consortium

J.C. Ambrose[23], P. Arumugam[23], E.L. Baple[23], M. Bleda[23], F. Boardman-Pretty[23,24], J.M. Boissiere[23], C.R. Boustred[23], H. Brittain[23], M.J. Caulfield[23,24], G.C. Chan[23], C.E.H. Craig[23], L.C. Daugherty[23], A. de Burca[23], A. Devereau[23], G. Elgar[23,24], R.E. Foulger[23], T. Fowler[23], P. Furió-Tarí[23], J.M. Hackett[23], D. Halai[23], A. Hamblin[23], S. Henderson[23,24], J.E. Holman[23], T.J.P. Hubbard[23], K. Ibáñez[23,24], R. Jackson[23], L.J. Jones[23,24], D. Kasperaviciute[23,24], M. Kayikci[23], L. Lahnstein[23], K. Lawson[23], S.E.A. Leigh[23], I.U.S. Leong[23], F.J. Lopez[23], F. Maleady-Crowe[23], J. Mason[23], E.M. McDonagh[23,24], L. Moutsianas[23,24], M. Mueller[23,24], N. Murugaesu[23], A.C. Need[23,24], C.A. Odhams[23], C. Patch[23,24], D. Perez-Gil[23], D. Polychronopoulos[23], J. Pullinger[23], T. Rahim[23], A. Rendon[23], P. Riesgo-Ferreiro[23], T. Rogers[23], M. Ryten[23], K. Savage[23], K. Sawant[23], R.H. Scott[23], A. Siddiq[23], A. Sieghart[23], K.R. Smith[23,24], A. Sosinsky[23,24], W. Spooner[23], H.E. Stevens[23], A. Stuckey[23], R. Sultana[23], E.R.A. Thomas[23,24], S.R. Thompson[23], C. Tregidgo[23], A. Tucci[23,24], E. Walsh[23], S.A. Watters[23], M.J. Welland[23], E. Williams[23], K. Witkowska[23,24], S.M. Wood[23,24] & M. Zarowiecki[23]

[23]Genomics England, London, UK. [24]William Harvey Research Institute, Queen Mary University of London, London EC1M 6BQ, UK

## The International Mouse Phenotyping Consortium

Susan Marschall[10], Christoph Lengger[10], Holger Maier[10], Claudia Seisenberger[14], Antje Bürger[14], Ralf Kühn[14], Joel Schick[14], Andreas Hörlein[14], Oskar Oritz[14], Florian Giesert[14], Joachim Beig[14], Janet Kenyon[9], Gemma Codner[9], Martin Fray[9], Sara J. Johnson[9], James Cleak[9], Zsombor Szoke-Kovacs[9], David Lafont[13], Valerie E. Vancollie[13], Robbie S.B. McLaren[13], Lena Hughes-Hallett[13], Christine Rowley[13], Emma Sanderson[13], Antonella Galli[13], Elizabeth Tuck[13], Angela Green[13], Catherine Tudor[13], Emma Siragher[13], Monika Dabrowska[13], Cecilia Icoresi Mazzeo[13], Mark Griffiths[13], David Gannon[13], Brendan Doe[13], Nicola Cockle[13], Andrea Kirton[13], Joanna Bottomley[13], Catherine Ingle[13], Edward Ryder[13], Diane Gleeson[13], Ramiro Ramirez-Solis[13], Marie-Christine Birling[19], Guillaume Pavlovic[19], Abdel Ayadi[19], Meziane Hamid[19], Ghina Bou About[19], Marie-France Champy[19], Hugues Jacobs[19], Olivia Wendling[19], Sophie Leblanc[19], Laurent Vasseur[19],

Elissa J. Chesler[3], Vivek Kumar[3], Jacqueline K. White[3], Karen L. Svenson[3], Jean-Paul Wiegand[3], Laura L. Anderson[3], Troy Wilcox[3], James Clark[3], Jennifer Ryan[3], James Denegre[3], Tim Stearns[3], Vivek Philip[3], Catherine Witmeyer[3], Lindsay Bates[3], Zachary Seavey[3], Pamela Stanley[3], Amelia Willet[3], Willson Roper[3], Julie Creed[3], Michayla Moore[3], Alex Dorr[3], Pamelia Fraungruber[3], Rose Presby[3], Matthew Mckay[3], Dong Nguyen-Bresinsky[3], Leslie Goodwin[3], Rachel Urban[3] & Coleen Kane[3]

