## [Peer Review File · Nature Communications]

Reviewers' Comments:

Reviewer #1:

Remarks to the Author:

This study presents a novel method for combining functional and conservation data to find essential genes and potentially improve genomic variant interpretation. By combining mouse and human cell line data with human constraint and disease information, the authors are able to categorise genes into broad categories relating to essentiality. The findings are novel and will be of general interest to the genomics community.

I have a few suggestions for improvement of the manuscript.

Results:

- Table 1: please give more detail. Are the genes mutually exclusive in each category? How many human genes are not represented at all? Is "number of genes" the number of human protein-coding genes? Given most human genes are not associated with disease, why are so few genes in the VN category?
- How is significance defined in the GO category enrichment (Fig 1b and Supp Table 2)?
- Section describing Fig 2c-e needs references for the statements regarding enrichment of essential genes in protein complexes and number of paralogues.
- The analysis of several DL genes to find novel diagnoses was interesting. Please detail in the text what variant consequences were included. Please also detail which DDD dataset was used and from how many probands the list of DNMs was obtained. This section could also be strengthened by the addition of more cases. Have the authors tried looking in <http://denovo-db.gs.washington.edu/denovo-db/>? Can GeneMatcher be used to find more cases?
- For the DDD case with a de novo mutation in TMEM63B: have the authors attempted to contact the clinician to get more detailed phenotypes, rather than simply using the "high-level phenotypes" available through the DECIPHER research track? This could potentially strengthen the phenotypic similarity with the 100KGP cases. It would also allow the authors to find out whether the child already has a diagnosis, or whether (if validated) this finding is likely to be the answer.

Discussion:

- What is the authors hypothesis for why CL genes are under-represented in disease classes? Are they embryonic lethal? Is there any evidence from the literature that can support/refute the hypothesis?
- In the 163 prioritised genes, some immediately jump out as potentially having similar roles to known developmental disorder genes based simply on gene names, e.g. CNOT4 (CNOT1 and 3 are known), PCGF1, (PCGF2 is known), SPTBN1 (SPTBN2 is known), etc. An analysis of the pathways or functional classes of these 163 genes compared with known monoallelic DD genes or the other DL genes would be interesting versus a control gene set such as the VN genes.

Figures:

- Fig 2a and 2i are almost unreadable; is there a better way to present these results? Wider violins perhaps?
- Fig 2i vs 3d; why are these in different figures? Suggest 3d is moved to Fig 2 so that Fig 3 is then focused only on disease data.
- Fig 3b and 3c; what about PanelApp genes? They all have MOI as well as organ/disease-specific information. Are the results the same?

- Fig 3g; better reference for DDG2P is now <https://www.ncbi.nlm.nih.gov/pubmed/31147538>
- Fig 3h (and corresponding text in Results); it is not clear what the "DDD resource" is, please explain.

Reviewer #2:

Remarks to the Author:

1. An excellent and systematic effort to add key functional insight for a large set of essential genes that contribute disproportionately to serious genetic disease (many developmental) in humans. The FUSIL categories are biologically and clinically meaningful. While not entirely novel—this is the team that has really been at the forefront of this type of categorization. Now it is all quite formal.

The translational vignettes (VPS4A and TMEM63B) are solid. Caveats well covered in the discussion. Overall an important timely contribution that has both basic scientific impact and significant clinical relevance.

2. Data, analysis into five function classes, integration of results with human data, and discussion are all top notch. Use and review of previous work is even-handed. Wonderful example of the use of IMPC results on viability of mouse mutants to the analysis and extension of rare mutations in humans.

3. Gorgeous figures and tables. Figures are clean and comprehensible almost without the legends. Lovely deep deep data.

4.

Minor comments (optional)

1. In abstract please avoid long noun phrases such as "human cell line essentiality screens"
2. "Regulate" versus "modulate" versus "control". In most situations gene variants modulate traits rather than regulating traits. Yes, the reader is a pedantic physiologist, but perhaps worth making the distinction.
3. This reader is a believer in Oxford comma. Not sure about Nature though ;-)
4. Third sentence of introduction is a bit of a monster and would be helped by a comma between the two major clauses. Better yet, two sentences—the first short and sweet; the second droning on but comprehensible.
5. Fourth sentence—all 47 words of it—also requires good breath control. Not all your readers are native English (or should I say German) readers. Introduction to introduce, not scare.
6. "Broad Institute Project Achilles": must we state the institution?
7. The sentence "The number of observed homozygous..." is ok grammatically, but nonetheless confusing to the reader. The problem is the two different uses of percentages in one sentence (one for animals, one for genes). Suggest dividing and conquering.
8. Third paragraph of introduction: would it make sense to tip hat to paralogue "buffering": thinking of HBB and peers, ACTC1 and peers.
9. "those genes where loss of function": "where" or "for which"
10. Ah finally, some active voice passion toward the end of the introduction. Could use a bit more of that in the first set of paragraphs.
11. Figure 1: Could definitely not mind the font sizes being bumped up where practical. Seems to be plenty of space on the left side of the figure and bottom too.
12. Page 10, perhaps this is the place for consideration of paralogues? Ah there it is in Figure 2e.
13. On the relation between recombination rate and gene category—please be clear in text and figure if you are referring only to human data (my impression). If so, is this replicated clearly in mouse?

14. Some readers may not know what GTE_x is. Specify species.

15. Page 13: "Whilst" really? (from <https://www.differencebetween.com/difference-between-while-and-vs-whilst/>) "Whilst is a word that has been used as a synonym of while for quite some time in English language. In fact, whilst is considered a Middle English word, whereas while is a much older word. Whilst is considered archaic and used more by poets and novelists. However, it is still used by people though some term it as old fashioned much like amongst in place of among and amidst in place of amid. When it comes to formal writing, whilst is preferred over while by the writers. It boils down to personal preference when choosing between while and whilst and you cannot be considered incorrect for making use of whilst in place of while."

16. Capitalization "network of Centers"

Reviewer #3:

Remarks to the Author:

The manuscript by Cacheiro and colleagues describes analysis of genes assessed by human cell line and mouse knockout essentiality. One major finding is that genes that are viable in human cell line knockout but lethal in mouse knockout are enriched with developmental disorder risk genes based on disease data and supported by pathway enrichment. On the other hand, the genes are lethal in both experiments are less enriched with known developmental disorder genes.

Major comments:

1. As a resource for disease gene discovery, the number of genes with information is really small. There are existing effective methods for the same purpose but cover all or nearly all genes, such as gnomAD constraint scores or other metrics based on large-scale functional genomics data. Since this is the main objective of the work, it is important to show the utility of this method in comparison with existing methods for gene discovery or its utility combined with existing methods.
2. I'm not convinced about the interpretation of pathway analysis presented in Figure 1. For example, CL is enriched with RNA-processing genes. It's well known that many RNA-processing genes are involved in cell differentiation and developmental processes in general. But only DL is enriched with developmental processes. It's not clear how many of RNA-process genes are included in the curated pathways about developmental processes. It would be helpful to clarify this point in pathway enrichment to make it more objective.

Minor comments:

1. Figure 2b: TPM is more naturally shown in log scale.
2. Figure 2i: pLI has bi-modal distribution (modes close to 0 and 1). The thin violin plot is not optimal to show the trend.

#####

Reviewers' comments:

#####

Reviewer #1 (Remarks to the Author):

This study presents a novel method for combining functional and conservation data to find essential genes and potentially improve genomic variant interpretation. By combining mouse and human cell line data with human constraint and disease information, the authors are able to categorise genes into broad categories relating to essentiality. The findings are novel and will be of general interest to the genomics community.

I have a few suggestions for improvement of the manuscript.

Results:

- Table 1: please give more detail. Are the genes mutually exclusive in each category? How many human genes are not represented at all? Is "number of genes" the number of human protein-coding genes? Given most human genes are not associated with disease, why are so few genes in the VN category?

We have lengthened the legend to Table 1 to add details that were previously only described in the methods. The genes in each category are mutually exclusive as indicated in the legend .

In terms of how many human genes are not represented, the number of human genes that are included in our study is limited by the number with human cell viability data (virtually all with 17,634 genes described), how many have a high-quality orthologue and how many of these orthologues have IMPC viability data (release 9.1 covers 4,934 genes) leaving a final number of 4,446 genes. This is explained in the Methods section of the paper and, as suggested by the Reviewer, we now include this information in the legend of Table 1: “for 4,446 protein-coding genes that have data in both resources and a high-quality orthologue”

We now clarify in the legend that the “number of genes” refers to human protein-coding genes: “and the number of human protein-coding genes is shown for each”.

Further detail has been added to the legend to describe how we define the viable with (VP) and without (VN) phenotype categories. Regarding why there are comparatively few genes in the VN category given most genes are not associated with (Mendelian) disease, it should be considered that a deviation detected in any one of 163 measured parameters will lead to the VP classification and not necessarily represent a Mendelian disease phenotype. Within this phenotypic spectrum we can find abnormal phenotypes (e.g. decreased heart rate, abnormal circulating glucose level) that may be associated to predisposition to common disease or even common phenotypic variation (e.g.

short/long tail) but do not necessarily represent Mendelian phenotypes, and we are focusing our analysis exclusively in genes associated to Mendelian disorders.

- How is significance defined in the GO category enrichment (Fig 1b and Supp Table 2)?

The Benjamini and Hochberg method (Journal of the Royal Statistical Society Series B-Statistical Methodology, 1995) was applied to correct for multiple testing and an adjusted P-value < 0.05 was considered significant. We now make this clearer throughout, as suggested by the Reviewer.

We have added more detail in the Methods section. We changed “BH method was applied for multiple testing correction and “In Fig. 1b, bubble size is proportional to the frequency of the term in the database and the colour indicates significance level as obtained in the enrichment analysis” to read:

“The Benjamin and Hochberg (1995) method was applied for multiple testing correction⁶⁴ and an adjusted P-value < 0.05 was considered significant.” and “ In Fig. 1b, bubble size is proportional to the frequency of the term in the database and the colour indicates the significance level as obtained in the enrichment analysis, after correcting for multiple testing.”

We added the following underlined text to the legend of Fig. 1b:

***“b) Gene Ontology Biological Process (GO BP) enrichment results.** Significantly enriched GO terms at the biological process level were computed using the set of IMPC mouse-to-human orthologues incorporated into the FUSIL categories as a reference (Table 1) and identified after correcting for multiple comparisons.”*

We have also included the relevant details the legends of some Supplementary Tables (Supplementary Tables 3, 4, 6, 7).

- Section describing Fig 2c-e needs references for the statements regarding enrichment of essential genes in protein complexes and number of paralogues.

We have now added reference tackling essential genes from the perspective of cell screens, sequence and links to Mendelian disease: “Similar continuous trends were observed for other gene features previously associated with essential genes^{17,31,39}, including protein-protein interaction network properties (Fig. 2c) or the likelihood of the gene product being part of a protein complex (Fig. 2d). CL genes also stand out as a singular category regarding the number of paralogues (Fig. 2e)”

- The analysis of several DL genes to find novel diagnoses was interesting. Please detail in the text what variant consequences were included. Please also detail which DDD dataset was used and from how many probands the list of DNMs was obtained. This section could also be strengthened by the addition of more cases.

**Have the authors tried looking in <http://denovo-db.gs.washington.edu/denovo-db/>?
Can GeneMatcher be used to find more cases?**

We have now incorporated the information about the type of variants that were investigated in Methods: for DDD functional de novo and rare LoF homozygous, compound heterozygous and hemizygous variants were queried and for the 100KGP, de novo variants predicted to have effect on the coding region of genes were queried.

We also added information on the DDD research variants dataset used in the same Methods section, including number of probands:

“This is a set of variants of unknown significance found in 4,293 children with developmental disorders who participated in the UK DDD study. It includes functional de novo and rare LoF homozygous, compound heterozygous and hemizygous variants in genes that are not associated to developmental disorders and are not reported in OMIM in children who remain undiagnosed in the DDD study”

We also updated the information in the main text:

“First, DDD makes publicly available a set of functional de novo variants and/or rare homozygous, hemizygous or compound heterozygous LoF research variants of unknown significance found in genes that are not associated with human disease according to OMIM and DDG2P. These variants were found in 4,293 children with developmental disorders who participated in the UK DDD study and remain undiagnosed”.

The suggestion to strengthen the novel diagnosis section by addition of more cases is very welcome and we were not previously aware of the denovo-db. Following the reviewer’s suggestion, we explored denovo-db database, containing a collection of germline de novo variants identified in the literature. We looked for the de novo coding variants in our set of 163 prioritised genes and we found denovo functional variants for 108 of them. These additional annotations are now included in Supplementary Table 10 and explained in the main text:

“We additionally explored denovo-db, a database containing de novo variants identified in the literature⁴⁴, and annotated our set of 163 prioritised genes with this information. We found at least one functional de novo variant reported in the database for 108 (66%) genes, of which 83 (51%) contain entries collected from resources other than DDD³ (Supplementary Table 10)”.

Interestingly, for VPS4A a nonsense (stop-gained) variant was found in a patient with Intellectual Disability (Lelieveld et al. 2016 - STable 2) and a missense variant in a patient with autism. None of these variants are present in gnomAD. We have now incorporated this information in the main manuscript, adding strength to this vignette, thanks to the reviewer’s suggestion:

“In addition, we found two de novo variants reported in denovo-db database⁴⁴: a nonsense (stop-gained) variant with an associated intellectual disability phenotype⁴⁷ and a missense variant in a patient with autism⁴⁸. None of these variants are present in gnomAD.”

[Redacted]

- For the DDD case with a de novo mutation in TMEM63B: have the authors attempted to contact the clinician to get more detailed phenotypes, rather than simply using the "high-level phenotypes" available through the DECIPHER research track? This could potentially strengthen the phenotypic similarity with the 100KGP cases. It would also allow the authors to find out whether the child already has a diagnosis, or whether (if validated) this finding is likely to be the answer.

We agree that more information on the detailed phenotypes for the DDD cases for TMEM63B (and other candidates) would potentially strengthen the case for disease association. We had already attempted to contact the recruiting clinicians via the DECIPHER/DDD team official route. Unfortunately, we were not successful in this regard. Candidates are only presented in the DDD research database for unsolved cases but if we had been able to make contact it is possible our candidates may have already been validated or rejected.

Discussion:

- What is the authors hypothesis for why CL genes are under-represented in disease classes? Are they embryonic lethal? Is there any evidence from the literature that can support/refute the hypothesis?

Indeed, this is one of the hypotheses we considered, and in the first paragraph of the Discussion, we mention: "In this model, highly essential genes will not be associated with human diseases because any function-altering mutation will likely lead to miscarriage or early embryonic death."

This set of genes corresponds to those genes found to be lethal both at the cellular and organismal level and, as we show in Figure 1c, are mostly associated with lethality in the mouse at a very early embryonic stage (prior to embryonic day E9.5).

We performed new analyses following the reviewer's suggestion (these are explained in the reply to the next comment). We show that CL genes are indeed more likely to interact with or to share pathways with known developmental genes than DL genes. So again, we hypothesise that the minimal enrichment for disease-associated genes in this category is due to an association to very early embryonic lethal phenotypes. We have also added a very recent reference that we were not aware of at the time of submission that supports the association of cellular lethal genes with lethality at early embryonic stages. Overall, this is covered with a new section in the first paragraph of the Discussion:

"The set of cellular lethal genes is indeed enriched for pathways associated to developmental disorders, but the minimal enrichment in developmental disease-

associated genes may be explained by embryonic lethality at a very early stage of development³⁶”.

- In the 163 prioritised genes, some immediately jump out as potentially having similar roles to known developmental disorder genes based simply on gene names, e.g. CNOT4 (CNOT1 and 3 are known), PCGF1, (PCGF2 is known), SPTBN1 (SPTBN2 is known), etc. An analysis of the pathways or functional classes of these 163 genes compared with known monoallelic DD genes or the other DL genes would be interesting versus a control gene set such as the VN genes.

This is a great suggestion and we have now performed several additional analyses, as suggested, incorporating pathways, protein families and protein-protein interaction information of known developmental disorder genes. We compared our set of 163 prioritised DL genes (not described to be associated to disease according to OMIM, Orphanet or DDG2P and highly intolerant to LOF) variation with non-disease associated genes from the other FUSIL bins in terms of likelihood of interacting with or sharing a protein family/biological pathway with known monoallelic developmental disorder genes. This is detailed in a new section in Methods:

“Further analysis was conducted to compare our set of 163 prioritised DL genes with those genes non associated to disease from the remaining FUSIL bins as well as with the entire set of genes in each FUSIL bin. We focused on evaluating our disease candidate genes against a set of genes associated to developmental disorders with a monoallelic mode of inheritance reported: the set of DDD monoallelic “green” genes as curated in Genomics England panelApp. We used protein family annotations from PFAM⁸⁸ (biomaRt, Ensembl Genes 97 version, hsapiens_gene_ensembl dataset [Data accessed 19.07.31])⁷⁴, pathways from Reactome⁶⁶ (lowest level pathways, Homo sapiens) [<https://reactome.org/download/current/Ensembl2Reactome.txt>; Data accessed 19.07.31] and protein interactors from STRING⁷⁶ as explained above (STRING pPI annotations with a combined score > 0.7). We compared our selected set of DL genes with those other in the different FUSIL bins that are not associated to disease and with the entire set of genes in each FUSIL bin. For each one of the three features (protein families, pathways and interactors), we computed the percentage of genes in each category sharing a PFAM protein family, a Reactome pathway (lowest level) or directly interacting with any monoallelic developmental disease gene”.

Results are shown in a new Supplementary Figure 5. We show that the prioritised set of 163 (non-disease associated) DL genes showed much higher percentages of genes than the non-prioritised set of (non-disease associated) DL genes sharing protein family, a Reactome pathway and protein interactions with a monoallelic developmental disease genes (according to the Genomics England DDD gene panel. This suggest that the prioritisation method is effective. Indeed, our set of candidate genes show the highest percentage of genes, except for (non-disease associated) CL genes for pathways and protein interactions, which were equal or marginally superior. These new results have been incorporated into the main text:

“These 163 prioritised DL genes are more likely to belong to the same protein family or biological pathway of known monoallelic developmental disease genes and more frequently interact with them when compared to genes across the FUSIL bins not associated to disease (Supplementary Fig. 5, Supplementary Table 9). Only the set of non-disease CL genes showed similar results regarding pathways and interactors.”

This latter observation, together with the overall lack of overrepresentation of disease genes in the CL category and the overlap of these genes with genes associated with early embryonic lethality in IMPC viability assessments, is likely explained by these genes acting early during embryonic development.

Additionally, we have added the information about the known monoallelic developmental genes that directly interact with, shared a protein family or a biological pathway with our set of 163 prioritised genes (Supplementary Table 9) for researchers wishing to explore our candidates in the future.

Figures:

- Fig 2a and 2i are almost unreadable; is there a better way to present these results? Wider violins perhaps?

We agree the readability of several plots was indeed difficult and have now replaced the violin plots by notched boxplots which hopefully facilitates the interpretation of the differences in the distribution between FUSIL categories for the different features, given that the CI around the median displayed with this visualization is an approximation of the evidence of median values differing between groups.

- Fig 2i vs 3d; why are these in different figures? Suggest 3d is moved to Fig 2 so that Fig 3 is then focused only on disease data.

We thank the reviewer for this suggestion and agree making this change has improved the logical presentation of our data. We also added a density plot of pLI scores to Figure 2 to better reflect the bimodal distribution of this metric.

- Fig 3b and 3c; what about PanelApp genes? They all have MOI as well as organ/disease-specific information. Are the results the same?

This is another welcome suggestion and we have now compared our results with those obtained when using PanelApp information.

These new analyses are now explained in the Methods section: “Given that OMIM contains a certain number of genes involved in susceptibility to multifactorial disorders and other non-mendelian gene-disease associations and that DDD-DDG2P also includes probable and possible gene-disease associations, we decided to investigate additional sets of curated gene-disease associations. In particular, we explored the gene panels curated by Genomics England and incorporated in its PanelAPP. Only those genes categorised as “green”, i.e. there is a high level of evidence for the gene-disease association, and therefore are considered as diagnostic-grade, were explored. Five different sets of genes were analysed: one corresponding to the total set of “green”

genes included in any Genomics England gene panel (PanelApp, 285 panels with at least one gene classified as green) and those genes belonging to the following panels: DDG2P (Developmental disorders, set of “green” genes from DDG2P panel, which contains a subset of DDG2P genes with one of the following levels of evidence: Confirmed or both DD and IF), PD (set of “green” genes from the Paediatric disorders gene panel), ID (set of “green” genes from the Intellectual disability gene panel.) and FA (set of “green” genes from the fetal anomalies panel, which contains a subset of genes associated to developmental disorders developed by the PAGE study (Prenatal Assessment of Genomes and Exomes) with a confirmed disease confidence rating that underwent additional review and curation) [<https://panelapp.genomicsengland.co.uk> ; Data accessed 19.06.02]

Also the annotations regarding the mode of inheritance:

“Genomics England PanelAPP also contain information about the mode of inheritance. We restricted the analysis to those genes associated with one of the following modes of inheritance: monoallelic, biallelic or both with consistent annotation across different panels. [<https://panelapp.genomicsengland.co.uk>; Data accessed 19.06.02].”

The results are shown in Supplementary Figure 4.

Results using GEL PanelAPP “green” genes and the MoI reported are very consistent with those using the OMIM-ORPHANET dataset. The enrichment in disease genes among the DL category is even higher when we used the curated and reviewed set of genes from DDG2P and the set of foetal anomalies associated genes from the PAGE Consortium. The following paragraph has been added to the results section:

“These findings were corroborated when we explored diagnostic-grade genes that have a high level of evidence for disease associations as curated by experts for Genomics England (Supplementary Fig. 4). The DL category showed an even higher enrichment in developmental disease-associated genes for a subset of highly confident genes from DDG2P⁴ (4.4 fold-increase; Supplementary Fig. 4e) and as did a set of genes associated to foetal anomalies from the PAGE Consortium⁴² (4.9 fold-increase; Supplementary Fig. 4f).”

Also the results about the MoI associated to these new subsets of genes involved in developmental disorders:

“but for developmental disorders, as represented by DDG2P, the DL fraction contained the majority of genes, reaching a percentage close to 50% for monoallelic disease genes (Fig. 3g,3h).), and up to 57% when different curated subsets of genes involved in developmental disorders and foetal anomalies were investigated (Supplementary Fig. 4g, 4h)”

One avenue of future research will focus on establishing collaborations with groups / consortia working on prenatal exome/genome sequencing.

- Fig 3g; better reference for DDG2P is now

<https://www.ncbi.nlm.nih.gov/pubmed/31147538>

We updated the reference for DDG2P throughout the manuscript.

- Fig 3h (and corresponding text in Results); it is not clear what the "DDD resource" is, please explain.

We added the following explanatory sentence to the main text: "This database reports genes that are likely to be causative of developmental disorders". We also indicate what DDD stands for in the legend of Fig. 3, as this was missing: "DDD/DDD-DDG2P, Deciphering Developmental Disorders database of genes that are likely causative of developmental disorders" as we agree this is key to understanding the Figure.

#####

Reviewer #2 (Remarks to the Author):

1. An excellent and systematic effort to add key functional insight for a large set of essential genes that contribute disproportionately to serious genetic disease (many developmental) in humans. The FUSIL categories are biologically and clinically meaningful. While not entirely novel—this is the team that has really been at the forefront of this type of categorization. Now it is all quite formal.

The translational vignettes (VPS4A and TMEM63B) are solid. Caveats well covered in the discussion. Overall an important timely contribution that has both basic scientific impact and significant clinical relevance.

2. Data, analysis into five function classes, integration of results with human data, and discussion are all top notch. Use and review of previous work is even-handed. Wonderful example of the use of IMPC results on viability of mouse mutants to the analysis and extension of rare mutations in humans.

3. Gorgeous figures and tables. Figures are clean and comprehensible almost without the legends. Lovely deep deep data.

Thank you for these positive comments.

4.

Minor comments (optional)

1. In abstract please avoid long noun phrases such as " human cell line essentiality screens"

We changed "and human cell line essentiality screens" to "and essentiality screens carried out on human cell lines"

2. "Regulate" versus "modulate" versus "control". In most situations gene variants modulate traits rather than regulating traits. Yes, the reader is a pedantic physiologist, but perhaps worth making the distinction.

We changed regulate to modulate as suggested.

3. This reader is a believer in Oxford comma. Not sure about Nature though ;-)

We think this is largely a British versus American English distinction but not sure what the Nature policy is either and are very happy to change to Oxford comma throughout if that is recommended.

4. Third sentence of introduction is a bit of a monster and would be helped by a comma between the two major clauses. Better yet, two sentences—the first short and sweet; the second droning on but comprehensible.

Thank you, this is a good suggestion. We have changed this sentence to read

“However, the majority of cases remain unsolved. One complimentary approach has been to incorporate gene-level information metrics. These metrics can help to identify candidate variants in previously unknown disease genes, which are subsequently confirmed as causative in functional in vitro and in vivo studies.”

5. Fourth sentence—all 47 words of it—also requires good breath control. Not all your readers are native English (or should I say German) readers. Introduction to introduce, not scare.

Thank you, another good suggestion. We have changed this sentence to read

“Measures of genetic intolerance to functional variation represent one class of metrics that have been used to prioritise candidate disease genes where heterozygous, dominant effects are suspected. These metrics are based on whole-exome and genome sequencing data from broad populations of healthy individuals or cohorts affected by non-severe and non-paediatric disease⁶⁻⁸”.

6. "Broad Institute Project Achilles": must we state the institution?

We agree and have changed it to just Project Achilles throughout. The reference already describes the source of this project.

7. The sentence "The number of observed homozygous..." is ok grammatically, but nonetheless confusing to the reader. The problem is the two different uses of percentages in one sentence (one for animals, one for genes). Suggest dividing and conquering.

We have changed this sentence:

“The number of observed homozygous LoF mice generated from an intercross between heterozygous parents allows the categorisation of a gene as lethal (0% homozygotes), subviable (<12.5% homozygotes) or viable, with ~25%, ~10% and ~65% of genes classified as such, respectively²³.”

to:

“The number of observed homozygous LoF mice generated from an intercross between heterozygous parents allows the categorisation of a gene as lethal (0% homozygotes), subviable (<12.5% homozygotes) or viable. The proportion of genes in each category are ~25%, ~10% and ~65%, respectively^{20,24}.”

8. Third paragraph of introduction: would it make sense to tip hat to paralogue "buffering": thinking of HBB and peers, ACTC1 and peers.

We do cover paralogue buffering in the discussion where we discuss the CL fraction but we have also added a statement to the introduction as suggested by changing "compensatory mutation rescuing necessary function"²⁹ through mechanisms such as paralogue buffering^{17,30}.

9. "those genes where loss of function": "where" or "for which"

Changed to "for which". The sentence now reads:

"We explore the FUSIL categories that span genes from those necessary for cellular survival all the way to those genes for which loss-of-function has no detected phenotypic impact on complex organisms, and we demonstrate.."

10. Ah finally, some active voice passion toward the end of the introduction. Could use a bit more of that in the first set of paragraphs.

We are struggling to identify examples to make the previous paragraphs describing the background read in the active voice but would definitely welcome some style guidance.

11. Figure 1: Could definitely not mind the font sizes being bumped up where practical. Seems to be plenty of space on the left side of the figure and bottom too.

We have increased the size of the Sankey diagrams and the GO enrichment plots in Figure 1, we hope the readability is better now.

12. Page 10, perhaps this is the place for consideration of paralogues? Ah there it is in Figure 2e.

Paralogues are addressed on page 11, where Fig. 2e is presented:

"CL genes also stand out as a singular category regarding the number of paralogues (Fig. 2e)."

and on page 23:

"The strong enrichment of CL genes for presence in protein complexes and a lack of paralogues would suggest these genes should be particularly intolerant to damaging mutations with no functional compensation that has evolved to buffer critical cell processes^{17,54,55}."

13. On the relation between recombination rate and gene category—please be clear in text and figure if you are referring only to human data (my impression). If so, is this replicated clearly in mouse?

We indicate in Fig. 2a that this is human data.

We have added the word “human” to the section “Recombination Rates” in the Methods section and also in page 11, where we talk about recombination rate, to increase clarity.

Thus, in the main text, we changed:

“Essential genes have been shown to be located in regions with lower recombination rates.”

to:

“In humans, essential genes have been shown to be located in regions with lower recombination rates.”

In this study we have focused on human gene annotations, an analysis of how these results replicate in the mouse it’s something we would like to address in a future project.

14. Some readers may not know what GTEx is. Specify species.

On page 11, We changed:

“Higher expression values have also been previously associated with essential genes¹⁶ and here decreasing expression, as measured by median GTEx expression across the entire range of tissues and cell lines, is observed following most to least essential FUSIL bins (Fig. 2b).”

to:

“Higher expression values have also been previously associated with essential genes¹⁷. Here we show a decreasing trend in human gene expression levels from most to least essential FUSIL bins, as measured by median GTEx expression across a wide range of tissues and cell lines (Fig. 2b).”

We also specify species (human) in legend of Fig. 2, on pages 19 and 21, and in the Methods (Gene expression section) where GTEx data is mentioned.

15. Page 13: "Whilst" really? (from

<https://www.differencebetween.com/difference-between-while-and-vs-whilst/>

"Whilst is a word that has been used as a synonym of while for quite some time in English language. In fact, whilst is considered a Middle English word, whereas while is a much older word. Whilst is considered archaic and used more by poets and novelists. However, it is still used by people though some term it as old fashioned much like amongst in place of among and amidst in place of amid. When it comes to formal writing, whilst is preferred over while by the writers. It boils down to personal preference when choosing between while and whilst and you cannot be considered incorrect for making use of whilst in place of while."

We have now changed to While throughout to be consistent.

16. Capitalization" "network of Centers"

Changed to "network of centers".

#####

Reviewer #3 (Remarks to the Author):

The manuscript by Cacheiro and colleagues describes analysis of genes assessed by human cell line and mouse knockout essentiality. One major finding is that genes that are viable in human cell line knockout but lethal in mouse knockout are enriched with developmental disorder risk genes based on disease data and supported by pathway enrichment. On the other hand, the genes are lethal in both experiments are less enriched with known developmental disorder genes.

Major comments:

1. As a resource for disease gene discovery, the number of genes with information is really small. There are existing effective methods for the same purpose but cover all or nearly all genes, such as gnomAD constraint scores or other metrics based on large-scale functional genomics data. Since this is the main objective of the work, it is important to show the utility of this method in comparison with existing methods for gene discovery or its utility combined with existing methods.

We agree that a current limitation is the number of genes with information, largely due to the starting set of IMPC genes with data, and we state this in the second to last paragraph of the discussion: “ Cross-species data integration is not without its limitations ... a significant proportion of the mouse protein-coding genome is not yet phenotyped by the IMPC and is lacking viability data”. The IMPC is a large-scale project, in its 8th year now, which is continuously generating new data with the aim of eventually covering all protein-coding genes and we aim to continuously update the resources presented in this paper on our portal.

However, the aim here is to provide a high-quality resource for disease gene discovery and therefore, as the reviewer states, is it critical that we demonstrate improved benefit. Rather than suggesting our framework as a replacement for already effective methods such as gnomAD constraint scores, we are proposing it as a complementary approach that aids gene prioritization when mouse data is available. In Fig 2i and 2j we demonstrate the correlation between specific FUSIL categories and constraint scores, in Figure 3 we show the utility of FUSIL categories for disease gene discovery in different contexts and, finally, in the prioritisation of the 163 novel gene candidates for developmental disorders we bring the two together by selecting DL genes that are also constrained by gnomAD scores. We realise, thanks to this helpful comment, that the power of combining our FUSIL framework with gnomAD constraint scores was not sufficiently explained. To address this we have now introduced a new Supplementary Figure 9 and text to explain the findings.

The additional benefit of combining gnomAD constraint scores with FUSIL categories to identify disease genes is shown for OMIM-Orphanet (compare Sup Fig. 9a and Sup Fig. 9b to Fig. 3a, Fig. 3g and Fig. 3h) and DDG2P (compare Sup Fig. 9c and Sup Fig. 9d with Fig. 3f, Fig. 3g and Fig. 3h) disease genes. Regarding the DL fraction, disease genes that are known to be intolerant to LoF according to pLI score are more enriched than the entire set of disease genes (OR 4.9 versus OR 2.6 for OMIM-Orphanet genes; OR 6.4 versus OR 3.9 for DDG2P genes). In addition, Sup Fig. 9b shows that

combining approaches increases the proportion of OMIM-Orphanet disease genes seen in the DL fraction e.g. from 37% to 50% for dominant diseases (see Fig. 3h). Sup Fig. 9d shows the same for DDG2P.

On a similar note we also demonstrate that combining methods improves over the performance of gnomAD scores alone. We compared the distribution of disease genes across pLI constraint categories (again using the suggested threshold of pLI > 0.90 to identify intolerant to LoF genes), particularly for monoallelic disease genes. While 43% of AD OMIM-Orphanet genes are highly constrained (Sup Fig. 9e), these genes are also more likely to occur in the 2 lethal FUSIL bins, so combining DL and pLi > 0.90 increases the recall of AD disease genes to 60% (Sup Fig. 9f). Once again, a similar trend is observed for the DDD-DDG2P dataset, with highly constrained disease genes more likely to occur in the 2 lethal FUSIL bins, CL and DL (Sup Fig. 9g, Sup Fig. 9h).

An additional feature offered by the FUSIL framework is the ability to utilise all the bins to focus on other types of disease such as late onset or recessive.

We have now covered this issue in the discussion section: “Combining approaches to compute intolerance to loss-of-function, e.g. integrating FUSIL bins with other constraint scores, improves the ability to identify disease-associated genes compared to the performance of standalone metrics (Supplementary Fig. 9). Moreover, although in the present study we have targeted a particular category of disorders, an additional feature offered by the FUSIL framework is the ability to utilise all the bins to focus on other types of disease, including those associated to late-onset, less severe phenotypes or recessive mode of inheritance. Alternative approaches integrating different strategies of gene prioritisation are expected to follow, acting as an accelerator in Mendelian disease-gene discovery.”

2. I’m not convinced about the interpretation of pathway analysis presented in Figure 1. For example, CL is enriched with RNA-processing genes. It’s well known that many RNA-processing genes are involved in cell differentiation and developmental processes in general. But only DL is enriched with developmental processes. It’s not clear how many of RNA-process genes are included in the curated pathways about developmental processes. It would be helpful to clarify this point in pathway enrichment to make it more objective.

We thank the reviewer for this observation and we have now performed an additional Reactome pathway analysis.

The IMPC is contributing to characterise genes that are generally poorly studied. We first used Gene Ontology biological process annotations as it provides better coverage by including many sources such as electronically inferred function based on homology as well as experimentally verified function described in the published literature. Using GO Biological Processes (BP), 91.36 % of genes have at least one functional annotation (56.7 % if we consider only annotations based on Experimental Evidence).

To better address the point raised by the reviewer, we performed a parallel analysis using curated pathways from Reactome as shown in the new Supplementary Figure 2 and Supplementary Table 4. As pathway genes tend to be highly studied and thus less

likely to be included in the IMPC dataset, our coverage was much lower with only 61.4 % of genes being associated with at least one pathway compared to the global GOBP coverage. Still, this new analysis confirmed our previous results indicating that CL genes are involved in basic cell functions such as cell cycle, DNA and multiple RNA and transcription associated pathways. We also saw that RNA-processing related pathways (RNA polymerase II transcription and generic transcription pathway) were nominally significantly enriched in the DL category (Supplementary Figure 2), however after correcting for multiple testing these pathways were no longer significantly enriched in the DL fraction (Supplementary Table 40). Several GO BP terms related to RNA polymerase were also overrepresented in the DL bin (Supplementary Table 3) as expected given that transcriptional regulation of gene expression is central to organism development as pointed out by the reviewer. In FUSIL we are simplifying a continuous feature, essentiality, into categories so some sharing or partial overlap of processes and pathways is not too surprising.

In summary, what we intended to highlight in this section is the fact that several morphogenic and developmental processes seem to be exclusively enriched in the DL category, supporting our approach to split the set of lethal genes in the mouse into two different categories. We have modified the text to describe the additional pathway analysis and highlight where both approaches agree and disagree, in particular for RNA processing:

“Whereas the set of CL genes was enriched for nuclear processes (DNA repair, chromosome organisation, regulation of nuclear division, among other cellular processes), the DL genes were enriched in morphogenesis and development functions (such as embryo development, appendage development, tissue morphogenesis and specification of symmetry). In contrast, genes in the SV and viable categories (VP, VN) were not significantly enriched in any biological process despite reasonable sample sizes, probably reflecting diverse roles for these genes. These results were consistent with those obtained in a pathway analysis, with genes belonging to cell cycle and DNA associated pathways significantly overrepresented among CL genes. Developmental biology pathways, on the other hand, were only found enriched among DL genes (Supplementary Fig 2, Supplementary Table 4). GOBP analysis showed numerous enriched RNA-processing related terms mainly in the CL category. Several related processes, particularly those involving RNA polymerase, were also enriched in the DL fraction, reflecting how transcriptional regulation is central to organism development. Taken together, these analyses provide additional evidence to make a distinction between the two sets of lethal genes in terms of their biological function

Minor comments:

1. Figure 2b: TPM is more naturally shown in log scale.

Thanks for pointing this out. We switched to a log scale for TPM values and also turned the violin plots into notched boxplots to facilitate interpretation of the distributions .

2. Figure 2i: pLI has bi-modal distribution (modes close to 0 and 1). The thin violin plot is not optimal to show the trend.

We have incorporated a density plot to reflect this bimodal distribution, where we can see that values close to 0 are more frequent in the two viable categories.

Reviewers' Comments:

Reviewer #1:

Remarks to the Author:

Thank you for you detailed responses and additional analyses. All the points raised in the previous round of review have been satisfactorily addressed.

Reviewer #3:

Remarks to the Author:

The authors have adequately addressed my previous comments.